# PRMT5 regulates alternative splicing of TCF3 under hypoxia to promote EMT and invasion in breast cancer

Srinivas Abhishek Mutnuru[1], Pooja Yadav[1¤], Parik Kakani[1], Shruti Ganesh Dhamdhere[1], Poorva Kumari[1], Shruti Agrawal[2], Atul Samaiya[3], Sanjeev Shukla[1]*

1 Department of Biological Sciences, Indian Institute of Science Education and Research Bhopal, Bhopal, Madhya Pradesh, India, 2 Department of Pathology, Bansal Hospital, Bhopal, Madhya Pradesh, India, 3 Department of Surgical Oncology, Bansal Hospital, Bhopal, Madhya Pradesh, India

¤ Current address: Beth Israel Deaconess Medical Center, Boston, Massachusetts, United States of America

* sanjeevs@iiserb.ac.in

## Abstract

Tumor hypoxia induced alterations in the epigenetic landscape and alternative splicing influence cellular adaptations. PRMT5 is a type II protein arginine methyltransferase that regulates several tumorigenic events in many cancer types. However, the regulation of PRMT5 and its direct implication on aberrant alternative splicing under hypoxia remains unexplored. In this study, we observed hypoxia-induced upregulation of PRMT5 via the CTCF in human breast cancer cells. Further, PRMT5-mediated symmetric arginine dimethylation H4R3me2s and H3R8me2s directly regulated the alternative splicing of *TCF3*. Under hypoxia, PRMT5-mediated histone dimethylation at the intronic conserved region (ICR) present between *TCF3* exon 18a and exon 18b recruits DNMT3A, resulting in DNA methylation. DNA methylation at the *TCF3*-ICR is recognized and bound by MeCP2 resulting in RNA-Pol II pausing, promoting the recruitment of the negative splicing factor PTBP1 to the splicing locus of *TCF3* pre-mRNA. PTBP1 promotes the exclusion of exon 18a which results in the production of the pro-invasive TCF3-18B (E47) isoform which promotes EMT and invasion of breast cancer cells under hypoxia. Collectively, our results indicate PRMT5-mediated symmetric arginine dimethylation of histones regulates alternative splicing of *TCF3* gene thereby enhancing EMT and invasion in breast cancer hypoxia.

## Introduction

Genesis of hypoxic niches in growing tumors triggers a wide array of changes in tumor cells allowing them to circumvent the adverse effects of lack of oxygen [1,2]. Epigenetic reprogramming and regulation of alternative splicing events

**Data availability statement:** All the data generated in this study are present in the paper and/or the Supporting information. All the replicate numerical values and uncropped blots that underlie the data summarized in Figs 1-6 and S2-S6 are provided in the supplementary information in the files S1 Data and S1 Raw Images. MCF7 ChIP seq data used in this study is publicly available at GEO database with the accession number GSE216843. The breast cancer patient microarray data from GSE76250 can be downloaded from https://xenabrowser.net/. RNA sequencing data of shControl and shPRMT5 cells performed in this study has been deposited at GEO database with the accession number GSE279474 and has been made public. This study does not report any original code.

**Funding:** S.A.M. is a recipient of funding from Indian Institute of Science Education and Research-Bhopal, India. P.Y. was supported by Council of Scientific and Industrial Research fellowship, India and is currently a post-doctoral researcher at BIDMC, Boston, MA, U.S.A. PA.K. is a recipient of postdoctoral fellowship from Indian Institute of Science Education and Research Bhopal, India, and Department of Biotechnology (DBT, India), Research Associate fellowship award. S.G.D is supported by Department of Biotechnology (DBT, India). PO.K is a graduate student at Indian Institute of Science Education and Research-Bhopal, India. This work was funded by a grant from (SERB) (https://serb.gov.in/)(CRG/2021/004949) and Department of Biotechnology (https://dbtepromis.nic.in/)(BT/PR44309/MED/30/2364/2021, BT/PR54659/BMS/85/494/2024) to SS. The funders did not play any role in the study design, data collection and analysis, decision to publish, or preparation of the manuscript.

**Competing interests:** The authors have declared that no competing interests exist.

**Abbreviations:** bHLH, basic-helix-loop-helix; EMT, epithelial–mesenchymal transition; gDNA, genomic DNA; GO, Gene Ontology; ICR, intronic conserved region; MeCP2, Methyl-CpG Binding Protein 2; MeDIP, Methylation-Dependent Immunoprecipitation; MXE, mutually exclusive exons; PBS, phosphate-buffered saline; PRMT5, protein arginine methyl transferase 5; qRT-PCR, quantitative RT-PCR; sgRNA, guide RNAs; shRNA, small hairpin RNA; TCF3, Transcription Factor 3.

facilitate adaptability of cancer cells to challenging conditions such as hypoxia [2–6]. Epigenetic alterations and aberrant alternative splicing have been independently studied and shown to aid tumor cell progression under hypoxia [5–13]. Considering the complexity and intricacies of alternative splicing, recent reports have suggested that chromatin structure, histone modifications, and epigenetic marks on DNA regulate alternative splicing outcomes [14–16]. The role of epigenetic modifications in regulating alternative splicing is a new dimension which is currently drawing significant attention and requires deeper and better understanding. Specifically, the regulation of such interplay between epigenetic modifications and alternative splicing in the context of hypoxia remains significantly understudied.

Protein arginine methyl transferase 5 (PRMT5) catalyzes the symmetric dimethylation of arginine residues of both histone and non-histone proteins [17,18]. PRMT5 has been reported to symmetrically dimethylate histones H3 and H4 at R8 and R3 positions, respectively [19,20]. PRMT5 has been observed to be upregulated in various cancer types and is known to regulate crucial tumorigenic events such as proliferation, EMT, invasion, and stem cell maintenance [21–23]. Additionally, under hypoxia, PRMT5 has been shown to regulate autophagy by symmetrically dimethylating ULK1 in LN229 (Glioblastoma), Huh7 (Hepatocellular carcinoma), and HOK (Human Oral Keratinocytes) cells [24]. Also, PRMT5 has been reported to be upregulated in HUVEC and A549 cells under ischemia induced hypoxic conditions regulating important cellular events such as angiogenesis and apoptosis [25,26]. These studies indicate that PRMT5 has a crucial role in regulating hypoxic stress response. However, hypoxia-mediated regulation of PRMT5 expression and its downstream effect on tumor progression in breast cancer models remain significantly less explored. Although, PRMT5 has been reported to affect gene transcription via histone methylation and pre-mRNA splicing via methylation of splicing factor proteins [27–29]. However, its specific role in regulating alternative splicing events via its ability to methylate histones has not yet been explored.

Consequently, our study aimed to address two notable gaps in the literature: (i) the mechanism of regulation of PRMT5 expression under hypoxia and (ii) the impact of PRMT5-mediated alternative splicing via histone methylation on tumor progression under hypoxia. By investigating the interplay between PRMT5-mediated histone modifications and alternative splicing in the context of hypoxia, we aimed to unravel the complexities associated with epigenetics-mediated regulation of alternative splicing under hypoxia in promoting tumor progression.

In this study, we have reported that PRMT5 is upregulated under hypoxia via CTCF, and its upregulation is essential for EMT and invasion of hypoxic breast cancer cells. Globally, we observed that loss of PRMT5 under hypoxia lead to a significant change in the alternative splicing landscape, and we identified *TCF3* as one of the novel alternative splicing targets of PRMT5. PRMT5 regulated *TCF3* alternative splicing involving exon 18A and exon 18B under hypoxia leading to the production of the pro-invasive *TCF3*-18B isoform. Mechanistically, we have

shown that, under hypoxia, PRMT5-mediated symmetric arginine dimethylation of histones H4 and H3 at R3 and R8 positions at an intronic conserved region (ICR) present between exon 18A and exon 18B of *TCF3* is directly responsible for the regulation of *TCF3* alternative splicing. Interestingly, we found that PRMT5-mediated histone methylation recruited DNMT3A to the splicing locus leading to DNA methylation of the *TCF3*-ICR region. Subsequently, we observed that binding of MeCP2 to the methylated DNA led to RNA pol II pausing under hypoxia which further resulted in the recruitment of the negative splicing factor PTBP1. PTBP1 helps in exon 18A exclusion under hypoxia resulting in the production of the *TCF3*-18B containing E47 isoform which helps tumor cell invasion under hypoxia. Notably, our study also addresses the intricate interplay between histone modifications, DNA methylation, RNA polymerase kinetics and splicing factor recruitment under hypoxia. This underlies the intimate dependency of chromatin modifications and splicing on each other and how they can be regulated in disease conditions to achieve adaptability. Remarkably, our findings for the first time, comprehensively demonstrate the nuanced relationship between PRMT5-mediated histone methylation, DNA methylation and alternative splicing governing tumor progression under hypoxia.

## Results

### PRMT5 is upregulated in breast cancer cells under hypoxia

PRMT5 has been reported to be upregulated in several tumor types [Kim and Ronai, 2020; Sapir and colleagues, 2021]. However, the influence of hypoxia upon *PRMT5* gene regulation remains unexplored. To address this caveat, we first analyzed a publicly available patient microarray data obtained from TCGA (GSE76250) where in the samples were stratified as hypoxia low or hypoxia high based on hypoxia signature gene expression [7]. Among all the PRMTs, we found *PRMT5* to be the most significantly upregulated gene under hypoxia. (Fig 1A and 1B). Additionally, we performed a Spearman's correlation test to observe the correlation of *PRMT5* gene expression with 10 classical hypoxia gene markers (*P4HA1*, *ENO2*, *NEDD4L*, *HK2*, *LDHA*, *FOXO3*, *SLC2A1*, *BHLHE40, PGGK1*, *BNIP3L*) using TCGA breast cancer tumor gene sets (BRCA Tumor). We observed a significant positive correlation ($R = 0.42$) between the gene expression of *PRMT5* and hypoxia markers (Fig 1C) indicating that hypoxia regulates *PRMT5* gene expression in human breast cancers. Next, to validate *PRMT5* expression under hypoxia, we performed quantitative RT-PCR (qRT-PCR) and immunoblotting to understand *PRMT5* gene and protein expression in two breast cancer cell lines MCF7 and MDA-MB-231 subjected to hypoxia. We found a significant upregulation in *PRMT5* gene (Fig 1D) and protein expression (Fig 1E) under hypoxia in both MCF7 and MDA-MB231 breast cancer cell lines. PRMT5 symmetrically dimethylates H3R8 and H4R3 [19,22,30]. Hence, we performed immunoblotting analysis to check the expression of PRMT5-mediated histone modifications H3R8me2s and H4R3me2s under hypoxia. We observed an increase in PRMT5-mediated histone marks H3R8me2s and H4R3me2s under hypoxia in both MCF7 and MDA-MB-231 cell lines (Fig 1E) indicating that PRMT5 upregulation could have functional implication via histone modifications under hypoxia. To extend our observations to clinical samples, we checked the *PRMT5* transcript and protein levels along with the PRMT5-mediated modifications H4R3me2s and H3R8me2s in patient-derived breast cancer cell line BC8322, subjected to hypoxia. We observed a significant increase in *PRMT5* mRNA (Fig 1F) as well as protein expression along with the corresponding increase in histone marks H4R3me2s and H3R8me2s under hypoxia (Fig 1G). Next, we performed IHC-F to detect PRMT5 expression in tumor sections derived from breast cancer patients. CA9 was used as a marker for hypoxia-positive tumor regions. Our immunostaining analysis showed a positive correspondence of CA9 and PRMT5 expression. Evidently, we found that regions with negative CA9 staining, indicative of non-hypoxic regions, showed reduced PRMT5 expression (Figs 1H and S1A). We also observed a significant positive correlation between PRMT5 and CA9 fluorescent staining indicating that PRMT5 is upregulated under hypoxia (Fig 1I) suggesting that PRMT5 upregulation is clinically relevant and can

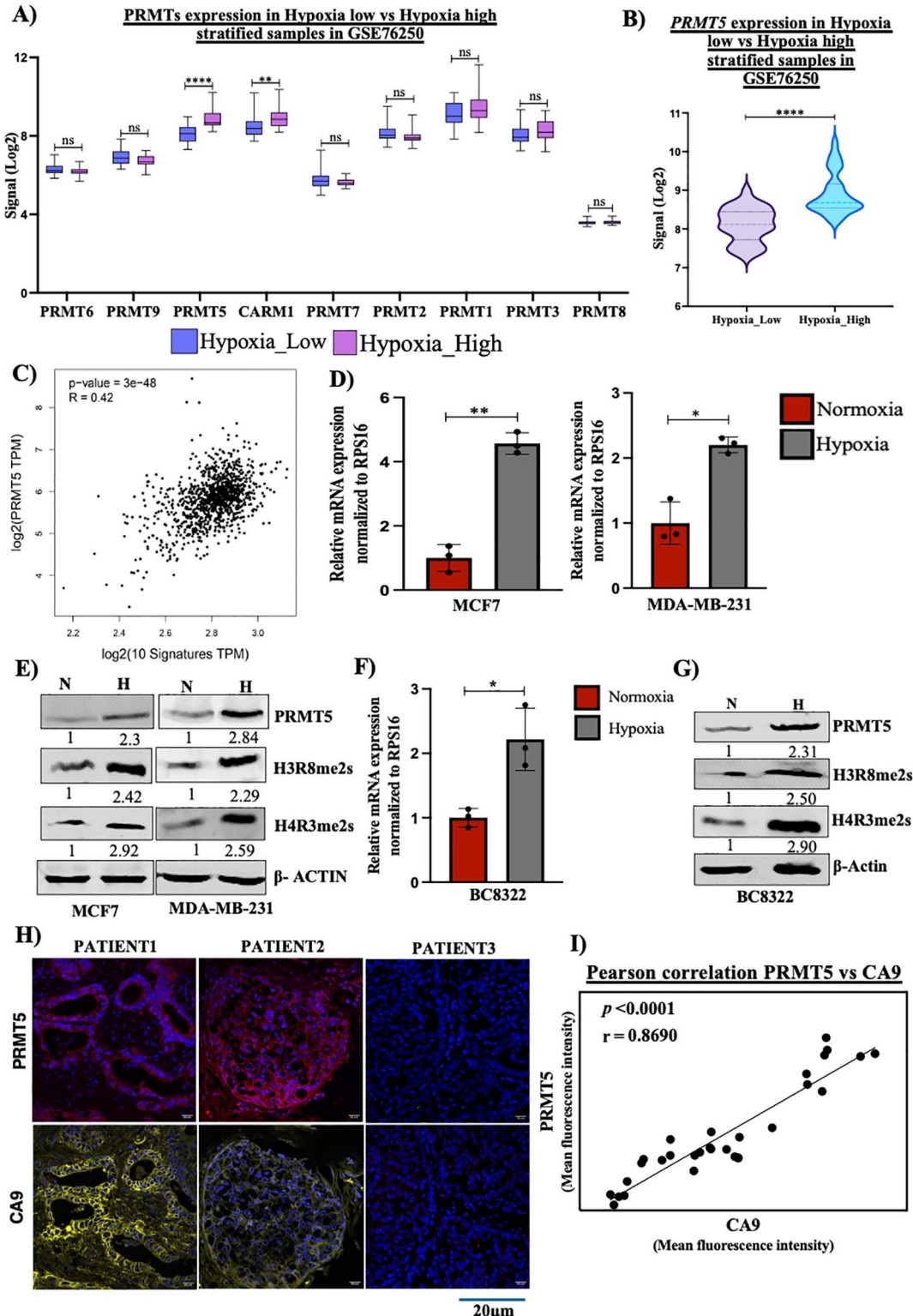

**Fig 1. PRMT5 is upregulated under hypoxia in breast cancer cells and clinical patient samples. A)** Graph showing Log2 signals of all PRMTs in breast cancer patient microarray data (GSE76520) stratified as Hypoxia low vs. Hypoxia high. **B)** PRMT5 mRNA expression in breast cancer patient microarray (GSE76520) stratified as Hypoxia low vs. Hypoxia high. **C)** Spearman's correlation analysis of PRMT5 mRNA expression with 10 hypoxia

marker genes. **D)** *PRMT5* mRNA expression in MCF7 and MDA-MB-231 breast cancer cell lines subjected to hypoxia. **E)** Immunoblot showing PRMT5, H4R3me2s, and H3R8me2s expression under hypoxia in MCF7 and MDA-MB-231 cell lines. **F)** *PRMT5* mRNA expression in patient derived cell line BC8322 subjected to hypoxia. **G)** Immunoblot showing PRMT5, H4R3me2s and H3R8me2s expression under hypoxia in in patient derived cell line BC8322. **H)** IHC-F analysis showing PRMT5 and CA9 expression in tumor sections of breast cancer patients. **I)** Pearson correlation analysis between PRMT5 and CA9 fluorescence intensities calculated from IHC-F. Scale bar 20 μm. Error bars, mean±SEM; two-tailed *t* test, *$p < 0.05$, **$p < 0.01$, $n = 3$ biological replicates. Numerical data of **(A–C)**, **(F)**, **(H)**, **(I)** available in S1 Data, sheet "*Figure 1*".

have implications in governing tumor progression. Together, these results indicated that PRMT5 is upregulated in breast cancer cells under hypoxia.

## CTCF is responsible for PRMT5 upregulation under hypoxia

Following our observation that PRMT5 is upregulated under hypoxia, we next wanted to identify the factors responsible for regulation of *PRMT5* expression under hypoxia. Since we observed a significant increase in *PRMT5* mRNA expression, we hypothesized that *PRMT5* is transcriptionally regulated under hypoxia. To understand the mechanism of transcriptional regulation of *PRMT5*, we generated 4 sequential promoter deletion constructs spanning the *PRMT5* promoter up to −2000 bp upstream of the TSS with 500 bp deletions in each construct. These promoter constructs were cloned upstream of firefly luciferase (Fluc) gene in a pGL3-basic vector (Fig 2A). Following this, we performed luciferase assay in both MCF7, and MDA-MB231 cells subjected to hypoxia. The luciferase construct harboring the full-length promoter (+100 to 2,000 bp) segment and the 1,500 bp segment (+100 to −1,500 bp) upstream of the transcription start site showed a significant increase in luciferase activity under hypoxia in both MCF7 and MDA-MB-231 cell lines (Figs 2B and S2A). This observation was indicative of the fact that a regulatory region present between −1,000 and 2,000 bp of the *PRMT5* promoter was responsible for regulating *PRMT5* expression under hypoxia. We then made use of publicly available ChIP-seq datasets from ChIP-Atlas to understand which transcription factors might potentially bind between the −1,000 and −2,000 bp region of the *PRMT5* promoter and regulate *PRMT5* expression under hypoxia. We identified a significant CTCF-binding site within the regulatory region (−1,000 to −2,000) at −1,175 bp position of the promoter which also had a significant binding score in the analysis performed in JASPAR (S2B and S2C Fig). Following this, we verified CTCF binding at *PRMT5* promoter under hypoxia using ChIP-Seq data in MCF7 cell line [31] (Fig 2C).

Firstly, to establish the role of CTCF in regulating *PRMT5* gene expression under hypoxia, we performed shRNA-mediated knockdown of CTCF in both MCF7 and MDA-MB-231 cells. Our immunoblotting analysis revealed a significant reduction in PRMT5 protein expression upon CTCF knockdown under hypoxia in both MCF7 and MDA-MB-231 cell lines (Figs 2D and S2D). DNA methylation is known to regulate gene expression [32–34] and since CTCF binding is also sensitive to DNA methylation [4,15,35], we performed Methylation-Dependent Immunoprecipitation (MeDIP)-qRT-PCR to analyze the DNA methylation profile at the CTCF-binding site of the *PRMT5* promoter (Fig 2E) to understand the role of promoter DNA methylation in regulating *PRMT5* expression under hypoxia. We observed a significant decrease in DNA methylation at the CTCF-binding site of the *PRMT5* promoter (Figs 2F and S2E) under hypoxia indicating that reduced DNA methylation under hypoxia could possibly allow the binding of CTCF to regulate *PRMT5* gene expression. Next, to confirm the binding of CTCF at the *PRMT5* promoter, we performed CTCF ChIP-qPCR assay and observed a significant increase in CTCF occupancy at the CTCF-binding site of the *PRMT5* promoter in both MCF7 and MDA-MB-231 cell lines (Figs 2G and S2F) suggesting that reduced DNA methylation at the CTCF-binding site allows CTCF binding on *PRMT5* promoter under hypoxia. In order to further verify the role of DNA methylation in regulating *PRMT5* expression under hypoxia, we made use of the dCAS9-DNMT3A epigenetic modifier system to preferentially methylate the CTCF-binding site of the *PRMT5* promoter (Fig 2H). The dCAS9-DNMT3A construct with the CTCF-binding site targeting sgRNA was transfected into MCF7 and MDA-MB-231

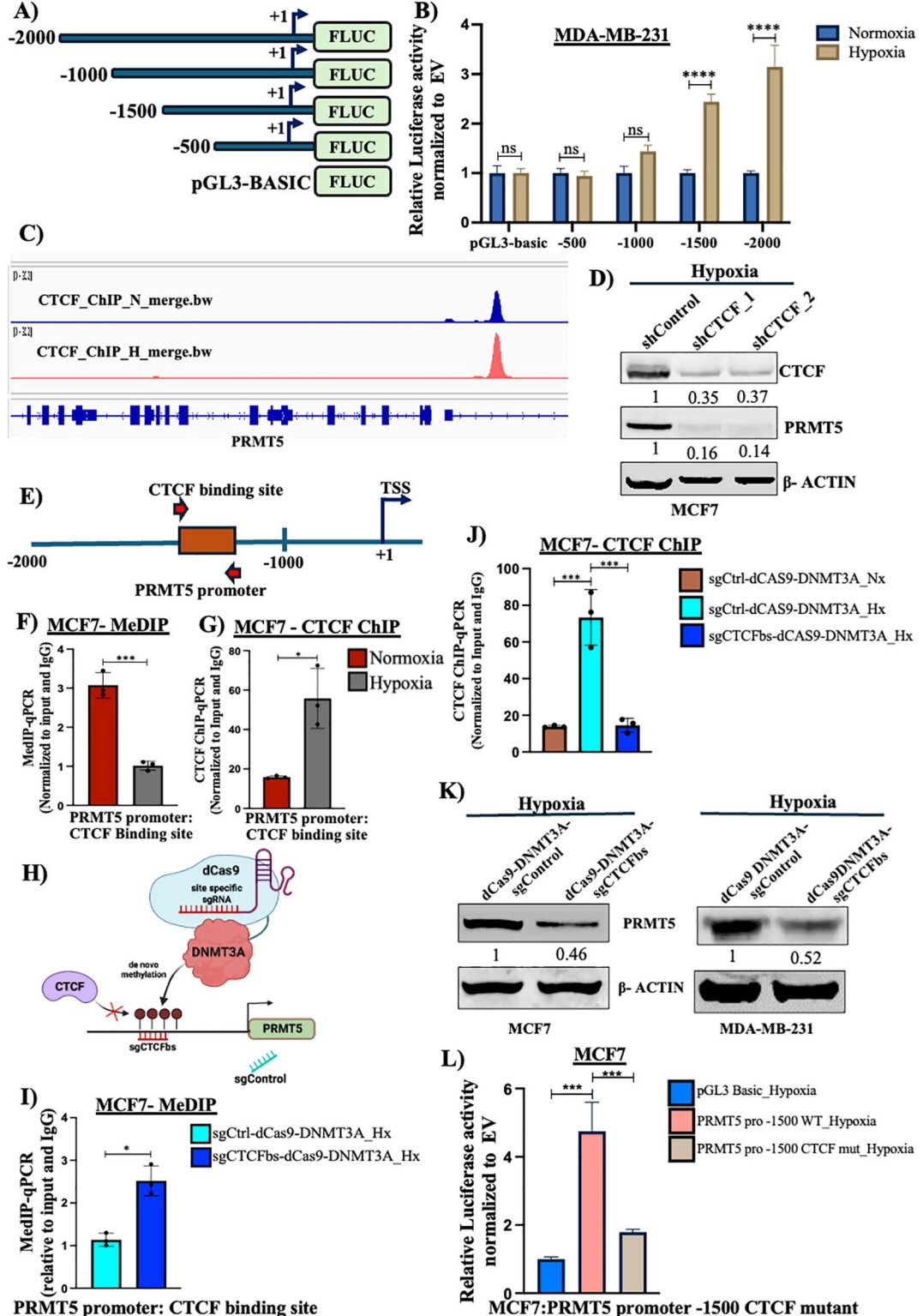

**Fig 2. CTCF is responsible for upregulation of PRMT5 under hypoxia in breast cancer cells. A)** Representation of PRMT5 promoter luciferase construct cloned upstream of F-Luc gene in pGL3 basic vector. **B)** Luciferase assay showing an increase in luciferase activity under hypoxia in MDA-MB-231 cells. **C)** Immunoblot showing decrease in PRMT5 expression upon CTCF KD under hypoxia in MCF7 cells. **D)** MCF7 CTCF ChIP-seq

track showing CTCF enrichment at PRMT5 promoter under hypoxia. **E)** Schematic representation of CTCF binding site at PRMT5 promoters along with the primer positions. **F)** MeDIP-qPCR showing decrease in DNA methylation at PRMT5 promoter in MCF7 cells normoxia vs. hypoxia. **G)** CTCF Chip qPCR showing enrichment in CTCF binding at PRMT5 promoter in MCF7 cells normoxia vs. hypoxia. **H)** Schematic representation of the dCAS9-DNMT3A epigenetic system targeting CTCF binding site of PRMT5 promoter. **I)** MeDIP qPCR showing enrichment of DNA methylation at PRMT5 promoter under hypoxia post dCAS9-DNMT3A sgCTCFbs vs. sgControl transfections. **J)** CTCF ChIP qPCR showing reduction in CTCF binding at PRMT5 promoter under hypoxia post dCAS9-DNMT3A sgCTCFbs vs. sgControl transfection. **K)** Immunoblot showing decrease in PRMT5 expression upon transfection with dCAS9-DNMT3A sgCTCFbs vs. sgControl. **L)** Luciferase assay showing decrease in luciferase activity in promoter construct harboring a mutated CTCF binding site. Error bars, mean ± SEM; two-tailed *t* test, one-way ANOVA. *$p < 0.05$, **$p < 0.01$, ***$p < 0.001$, ****$p < 0.0001$, $n = 3$ biological replicates. Numerical data of (B), (F–G), (I–J), (L) available in S1 Data, sheet *"Figure 2."* Fig 2H, Created in BioRender. Shukla, S. (2025) https://BioRender.com/74kzhl6.

cells under hypoxia following which we analyzed for DNA methylation profile, CTCF binding, and *PRMT5* expression. Firstly, we observed a significant increase in DNA methylation (Fig 2I) and a concomitant decrease in CTCF binding at the CTCF-binding site (Fig 2J) of the *PRMT5* promoter upon transfection with the dCAS9-DNMT3A construct targeting the CTCF-binding site of the *PRMT5* promoter under hypoxia. This suggested that decrease in methylation under hypoxia is necessary for CTCF binding at the *PRMT5* promoter. Expectantly, our immunoblot analysis revealed a significant decrease in PRMT5 protein expression (Fig 2K) indicating that under hypoxia, decrease in DNA methylation allows CTCF binding at the *PRMT5* promoter thereby regulating *PRMT5* expression. Furthermore, we mutated the CTCF-binding site present in the promoter-luciferase construct (+1 to −1,500 bp) (S2G Fig) and performed luciferase assay under hypoxia to establish the role of CTCF-mediated transcriptional upregulation of *PRMT5*. In corroboration with our previous results, we observed a significant reduction in luciferase activity in the construct harboring mutant CTCF binding site under hypoxia (Figs 2L and S2H) suggesting that CTCF is responsible for *PRMT5* upregulation under hypoxia. Put together, our observations suggested that reduction in DNA methylation at the CTCF-binding site of the *PRMT5* promoter allowed CTCF binding and regulation of *PRMT5* expression under hypoxia.

### PRMT5 regulates EMT and invasion of breast cancer cells under hypoxia

Metastasis is responsible for more than 90% of cancer-related mortalities [36–38]. Therefore, it becomes imperative to improve our understanding of the molecular mechanisms governing the complex process of metastatic cascade in order to tackle its adverse effects. Hypoxia is known to induce EMT and invasion of cancer cells [4]. However, there have been no reports with respect to the role of PRMT5 in regulation of EMT and invasion under hypoxia. Therefore, we envisaged to understand the role of PRMT5 in regulating EMT and invasion under hypoxia.

Firstly, we performed shRNA-mediated PRMT5 knockdown under hypoxia in both MCF7 and MDA-MB231 cell lines to check the effect of PRMT5 on EMT induction. We performed qRT-PCR to check the transcript levels of the mesenchymal marker genes *VIM* (Vimentin) and *CDH2* (N-Cadherin) along with the epithelial marker gene *CDH1* (E-Cadherin) and the EMT-regulating transcription factor *SNAI1* (Snail) in both MCF7 and MDA-MB-231 cell lines. We observed a significant decrease in the transcript levels of the mesenchymal markers *VIM* and *CDH2* along with EMT transcription factor *SNAI1* (Figs 3A and S3A). Epithelial marker *CDH1* showed a significant increase in transcript levels upon PRMT5 downregulation under hypoxia in both MCF7 and MDA-MB-231 cell lines (Figs 3A and S3A). We then analyzed the protein levels of epithelial markers E-cad and Cytokeratin-18 in MCF7, and Cytokeratin-18 in MDA-MB-231. Protein levels of mesenchymal marker Vimentin along with EMT-regulating transcription factor Snail were also analyzed in both MCF7 and MDA-MB-231 cell lines. Our immunoblot analyses showed an increase in the epithelial marker E-Cad and Cytokeratin 18 upon PRMT5 downregulation under hypoxia in both MCF7 and MDA-MB-231 cell lines whereas the mesenchymal marker Vimentin along with the transcription factor Snail showed decreased protein expression upon PRMT5 downregulation under hypoxia in both MCF7 and MDA-MB-231 cell lines (Figs 3B and S3B). Therefore, these results suggested that PRMT5 somehow

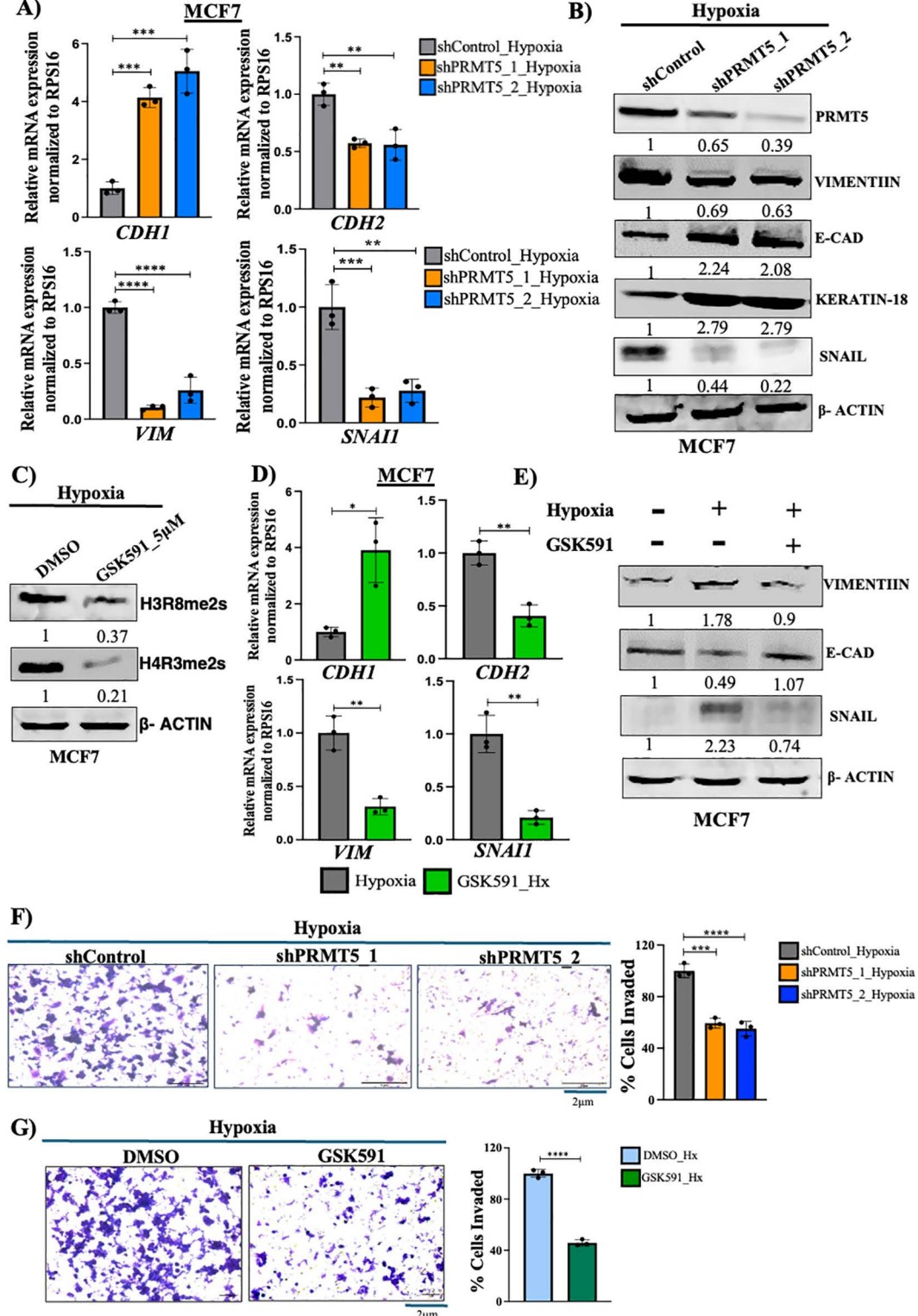

**Fig 3. PRMT5 regulates EMT and invasion under hypoxia. A)** qRT-PCR depicting mRNA expression of EMT markers in shCTRL vs. shPRMT5 MCF7 cells under hypoxia. **B)** Immunoblot showing protein level of EMT markers in shCTRL vs. shPRMT5 MCF7 cells under hypoxia. **C)** Immunoblot showing decrease in histone marks HR38me2s and H4R3me2s upon vehicle (DMSO) vs. 5 μM GSK591 treatment in MCF7 cells under hypoxia. **D)** qRT-PCR

depicting mRNA expression of EMT markers upon vehicle (DMSO) vs. 5 μM GSK591 treatment in MCF7 cells under hypoxia. **E)** Immunoblot showing protein level of EMT markers upon vehicle (DMSO) vs. 5 μM GSK591 treatment in MCF7 cells under hypoxia. **F)** Matrigel invasion assay and its respective quantification (right side) in shCTRL vs. shPRMT5 PRMT5 KD MCF7 cells under hypoxia. **G)** Matrigel invasion assay and its respective quantification (right side) upon vehicle (DMSO) vs. 5 μM GSK591 treatment in MCF7 cells under hypoxia. Scale bar 2 μm. Error bars, mean±SEM; two-tailed $t$ test, one-way ANOVA. *$p < 0.05$, **$p < 0.01$, ***$p < 0.001$, ****$p < 0.0001$, $n = 3$ biological replicates. Numerical data of (A), (D), (F–G) available in S1 Data, sheet "*Figure 3*".

regulated EMT induction under hypoxia. As observed earlier, PRMT5-mediated histone modifications H3R8me2s and H4R3me2s were found to be upregulated under hypoxia. Therefore, we hypothesized if obliterating these histone modifications was sufficient to curb EMT induction under hypoxia. To test this, we used GSK591 which is a substrate competitive inhibitor of PRMT5 that exerts its effect by blocking the ability of PRMT5-MEP50 complex to methylate target proteins [39,40]. We treated MCF7 and MDAMB-231 cells with 5 μM GSK591 under hypoxia [21] and found a significant reduction in the histone marks H3R8me2s and H4R3me2s (Figs 3C and S3C) indicating that GSK591 specifically inhibits PRMT5 activity. Along with this, as observed earlier, our q-RT-PCR results showed a significant decrease in the transcript levels of the mesenchymal markers *VIM* and *CDH2* along with EMT transcription factor *SNAI1* whereas the Epithelial marker *CDH1* showed a significant increase in transcript levels (Figs 3D and S3D). We also found that the epithelial markers E-cad and Cytokeratin 18 showed an increase in protein expression upon PRMT5 activity inhibition under hypoxia, whereas the mesenchymal marker Vimentin along with the transcription factor Snail showed decreased protein expression upon PRMT5 activity inhibition under hypoxia in MCF7 and MDA-MB-231 cell lines (Figs 3E and S3E). Together, these observations suggested that PRMT5-mediated histone modifications (H3R8me2s/H4R3me2s) could be one of the major players involved in regulation of EMT induction under hypoxia.

Induction of EMT subsequently renders epithelial cells more invasive which initially invade through the adjacent basement membrane to initiate the first step of the metastatic cascade (Cao and colleagues, 2023). Therefore, we intended to study PRMT5-mediated acquisition of invasive potential of breast cancer cells under hypoxia. We performed Matrigel invasion assay and observed a significant decrease in the invasive potential of both MCF7 and MDA-MB-231 cell lines upon PRMT5 downregulation under hypoxia (Figs 3F and S3F). Additionally, treatment of MCF7 and MDA-MB-231 cell lines with 5 μM GSK591, similarly decreased the invasive potential of both breast cancer cell lines under hypoxia (Figs 3G and S3G). In summary, our findings demonstrated that PRMT5-mediated histone modifications could possibly regulate EMT and invasion of breast cancer cells under hypoxia.

## PRMT5 regulates *TCF3* splicing under hypoxia

The primary goal of our study was to investigate the role of PRMT5-mediated regulation of alternative splicing via modulation of histone modifications. Since, we observed a positive correlation between reduction of PRMT5-mediated histone methylation and EMT induction upon GSK591 treatment under hypoxia, we speculated the involvement of PRMT5-mediated histone methylation in regulation of EMT. Our next step was to investigate how PRMT5-mediated histone methylation could regulate alternative splicing thereby impacting EMT induction under hypoxia.

Firstly, to understand the effect of PRMT5-mediated changes in alternative splicing events under hypoxia, we performed shRNA-mediated knockdown of PRMT5 under hypoxia in MDA-MB-231 cells followed by RNA-Seq. RNA-Seq analysis was performed and various splicing events were classified. Among the total 6,928 significant splicing events identified (FDR < 0.05 and |PSI| > 10%), 54% were cassette exon events (skipped exons), 32% were mutually exclusive exons (MXE), 4% were retained introns, and 4% and 6% were alternative 5′ splice site and alternative 3′ splice site selection events, respectively (Fig 4A). Mutually exclusive exon splicing events are often complex which produce functional transcripts of varying functions [41]. Unlike cassette exon events which can disrupt the protein sequence, MXE often

PLOS Biology

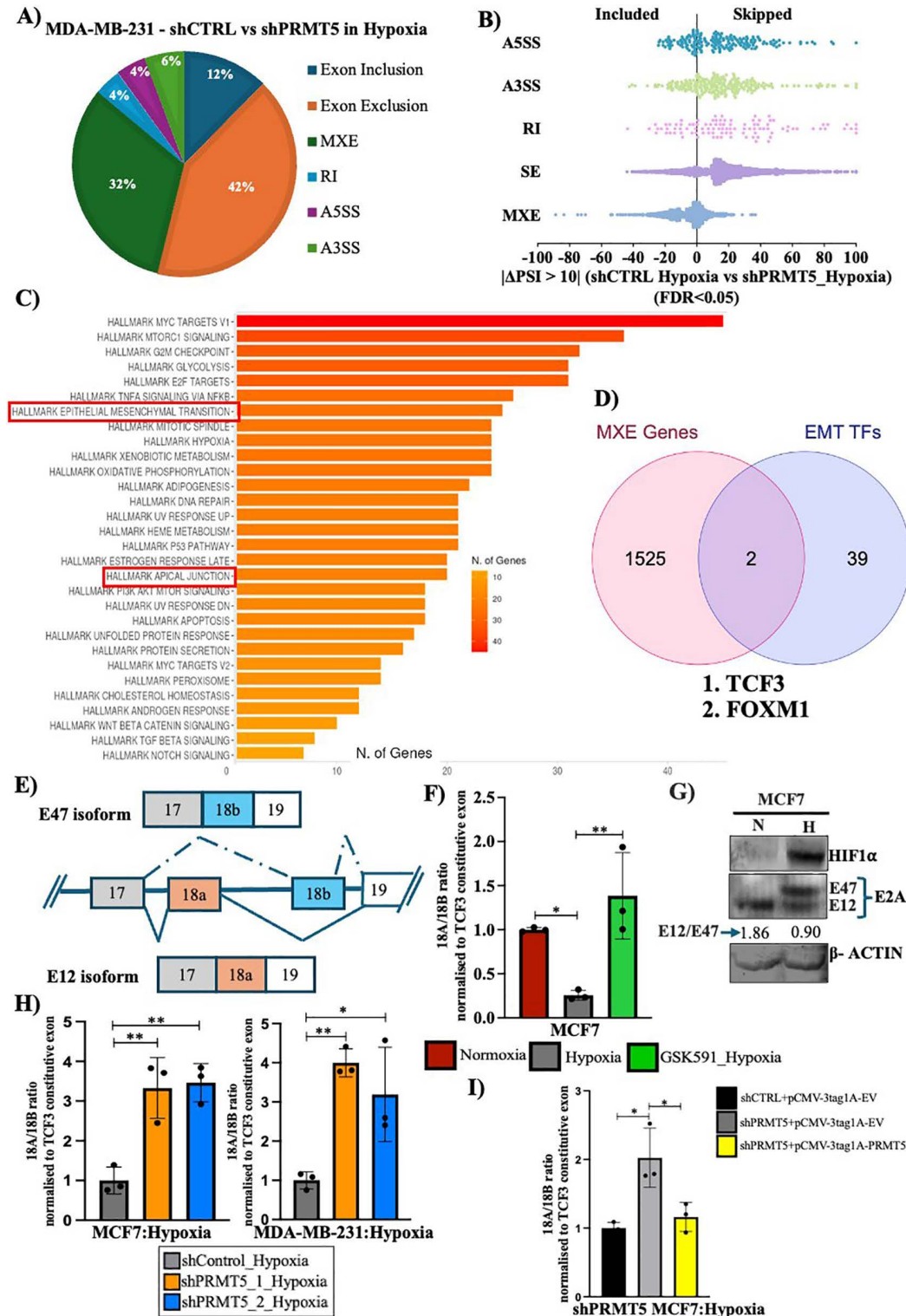

**Fig 4. PRMT5 regulates alternative splicing landscape under hypoxia. A)** Pie chart showing distribution of different types of significant AS events (FDR<0.05) in shCTRL vs. shPRMT5 MDA-MB-231 cells under hypoxia. **B)** Distribution of ΔPSI of significant AS events in shCTRL vs. shPRMT5 MDA-MB-231 cells under hypoxia. **C)** Gene ontology analysis of genes showing significant MXE events upon PRMT5 KD under hypoxia. **D)** Venn diagram showing *TCF3* and *FOXM1,* as novel alternatively spliced targets of PRMT5 obtained by overlapping EMT transcription factors gene set vs. gens

undergoing MXE event in shCTRL vs. shPRMT5 MDA-MB-231 cells under hypoxia. **E)** Schematic depiction of *TCF3* alternative splicing. **F)** qRT-PCR showing change in exon 18A/18B ratio upon vehicle (DMSO) vs. 5 µM GSK591 treatment in MCF7 cells. **G)** Immunoblot depicting protein expression of E12 and E47 isoform in MCF7 cells, normoxia vs. hypoxia. **H)** qRT-PCR showing an increase in exon 18A/18B ratio in shCTRL vs. shPRMT5 MCF7 and MDA-MB-231 cells under hypoxia. **I)** qRT-PCR showing a decrease in 18A/18B ratio upon PRMT5 OE in shPRMT5 MCF7 cells under hypoxia. Error bars, mean ± SEM; two-tailed *t* test, one-way ANOVA. *$p < 0.05$, **$p < 0.01$, ***$p < 0.001$, ****$p < 0.0001$, $n = 3$ biological replicates. Numerical data of (A–B) (F) (H–I) available in S1 Data, sheet "*Figure 4*".

results in diversification of protein function from a single transcript often implying context-dependent tight and complex regulation [42,43]. In the context of tumor progression, mutually exclusive splicing events such as *PKM* (metabolism) and *FGFR* (EMT) display how splice switching often favors aggressive tumor progression [14,16,44–46]. Therefore, in order to understand PRMT5-mediated regulation of alternative splicing in promoting EMT under hypoxia, we chose to further analyze MXE events (Fig 4B). Gene Ontology (GO) analysis of the genes undergoing MXE splicing events (FDR < 0.05 and |ΔPSI| > 10%) upon PRMT5 knockdown under hypoxia revealed regulation of apical junctions and epithelial–mesenchymal transition (EMT) to be among the top enriched categories (Fig 4C). This further suggested that PRMT5-mediated changes in alternative splicing significantly impacted EMT processes under hypoxia.

EMT transcription factors are critical regulators of EMT induction and are essential for the initiation and progression of cancer cells to a more aggressive and metastatic phenotype [47–49]. Therefore, we overlapped our dataset of genes showing MXE events with a dataset of EMT transcription factors [50] with the aim to identify EMT factors undergoing PRMT5-mediated alternative splicing under hypoxia. Among all the EMT-regulating transcription factors, TCF3 and FOXM1 showed significant splicing changes upon PRMT5 KD under hypoxia (Fig 4D). To delineate the detailed mechanism of how PRMT5 regulated alternative splicing under hypoxia, we further selected *TCF3* as our model gene system.

Transcription Factor 3 (TCF3) is a member of the basic-helix-loop-helix (bHLH) transcription factors that has been previously shown to repress E-cad expression in differentiated cells thereby promoting invasive phenotype [51]. Additionally, *TCF3* gene has been shown to promote cancer invasiveness and stemness in different cancer types [52–54]. So far, there have been no reports suggesting how *TCF3* splicing is affected in tumor hypoxia and its role in orchestrating EMT and invasion. Therefore, we hypothesized that PRMT5 regulates *TCF3* splicing via modulating the histone methylation thereby promoting EMT and invasion under hypoxia.

*TCF3* alternative splicing event involves exon 18A and exon 18B. Inclusion of exon 18A results in the production of the *TCF3*-18A (E12) isoform whereas, inclusion of 18B leads to the production of the proinvasive *TCF3*-18B (E47) isoform (Fig 4E) [51]. Our RNA-Seq analysis displayed increased inclusion of exon 18A (|ΔPSI| > 10% and FDR < 0.05) upon PRMT5 KD under hypoxia. Since these exons are mutually included in the final transcript, we wanted to analyze how exon 18A inclusion was regulated with respect to exon 18B under hypoxia. Firstly, to understand *TCF3* splicing pattern and its dependency on PRMT5 in breast cancer hypoxia, we performed qRT-PCR analysis with or without GSK591 treatment using primers designed specifically against exons 18A and 18B. We sequence verified the specificity of these primers and confirmed that they amplified the intended exons only. We observed that, under hypoxia, *TCF3* exon 18A/18B ratio was significantly downregulated but was rescued back upon PRMT5 activity inhibition by GSK591 indicating that exon 18A inclusion was significantly downregulated under hypoxia in both MCF7 and MDA-MB-231 cells and that PRMT5 was involved in exon 18A exclusion under hypoxia (Figs 4F and S4A). Additionally, the effect of GSK591-mediated inhibition of PRMT5 activity on TCF3 splicing was also an indication of the fact that PRMT5-mediated histone methylation could directly be involved in dictating TCF3 splicing outcomes. In order to verify the effect of splicing at the protein level, we performed immunoblotting to detect the change in expression of the TCF3 isoforms. We made use of an antibody which would detect the common region present in both the isoforms. Under hypoxia, we observed a decrease in the E12 (*TCF3*-18A) isoform with a concomitant increase in E47 (*TCF3*-18B) isoform under hypoxia (Fig 4G). This observation suggested that the splicing shift resulting in the increase of the pro-invasive E47 isoform under hypoxia was an important event and could have a decisive effect on the regulation of EMT and invasion under hypoxia. Next, we performed shRNA-mediated

knockdown of PRMT5 in both MCF7 and MDA-MB-231 cell lines under hypoxia to verify PRMT5-mediated regulation of *TCF3* splicing. Our qRT-PCR analysis showed an increase in exon 18A/18B ration upon PRMT5 knockdown under hypoxia (Fig 4H). To rescue this effect, we overexpressed PRMT5 in PRMT5 downregulated cells (S4B Fig) following which we performed qRT-PCR analysis to check if replenishing PRMT5 expression overturned the splicing event. As expected, overexpression of PRMT5 under hypoxia led to a significant decrease in the exon 18A/18B ratio indicating that PRMT5 restricted 18A inclusion under hypoxia (Fig 4I). Together, these results showed that PRMT5-mediated symmetric arginine histone dimethylation regulated *TCF3* splicing event under hypoxia by causing exon 18A exclusion.

## PRMT5-mediated H4R3me2s modification is essential and sufficient to regulate *TCF3* splicing under hypoxia

Until now, we delineated that PRMT5-mediated histone methylation regulated *TCF3* splicing under hypoxia. We next wanted to systematically deduce the mechanism of how PRMT5-mediated symmetric arginine dimethylation of histones affected the *TCF3* splicing outcome under hypoxia. We analyzed the *TCF3* splicing locus and identified an evolutionarily conserved region located in the intron between exon 18A and 18B (Fig 5A) which has been previously shown to be involved in TCF3 splicing event [55]. Therefore, we investigated if the ICR could potentially be a site which could be utilized by PRMT5 for exerting its effect on *TCF3* splicing regulation. To check this, we performed H3R8me2s and H4R3me2s ChIP-qPCR analysis and checked for the enrichment of these PRMT5-mediated histone marks at the *TCF3*-ICR region in both MCF7 and MDA-MB-231 cells under hypoxia. We observed a significant increase in both the histone marks, H3R8me2s and H4R3me2s at the *TCF3*-ICR region under hypoxia (Figs 5B and S4B). This result certainly was in corroboration with our previous observation that PRMT5-mediated histone methylation indeed played a role in dictating alternative splicing of *TCF3* under hypoxia. To establish that PRMT5-mediated histone methylation at the *TCF3*-ICR region is absolutely essential to regulate alternative splicing, we employed a dCAS9 epigenome modification system. We, for the first time generated a dCAS9-PRMT5 construct containing the full-length PRMT5 gene downstream of dCAS9 which can be used to regulate symmetric dimethylation of histones at target regions (S4C and S4D Fig). We made use of this dCAS9-PRMT5 construct to target the *TCF3*-ICR region and checked the effect of local histone symmetric dimethylation event on *TCF3* splicing (Fig 5C). Firstly, we performed shRNA-mediated knockdown of PRMT5 to deplete endogenous PRMT5 protein following which the cells were transfected with the dCAS9-PRMT5 construct targeting the *TCF3*-ICR region to specifically modify the local histone symmetric dimethylation mark at the *TCF3* splicing locus. We then verified if the dCAS9-PRMT5 construct was able to induce symmetric dimethylation of histones at H4R3 and H3R8 positions at the desired position of the splicing locus. Our ChIP-qPCR analysis showed that targeting dCAS9-PRMT5 at the *TCF3*-ICR region was able to catalyze both H4R3me2s and H3R8me2s modifications at the *TCF3*-ICR region (Fig 5D). Having established that the dCAS9-PRMT5 construct was specifically able to methylate the histones at the target site, we then used the construct to check if local modifications of H4R3me2s and H3R8me2s at the *TCF3*-ICR region was sufficient to alter *TCF3* splicing event. We performed qRT-PCR analysis following dCAS9-PRMT5 transfection in PRMT5 downregulated cells to check the change in *TCF3* alternative splicing pattern. We observed a significant decrease in the 18A/18B ratio upon targeting dCAS9-PRMT5 to the *TCF3* splicing locus (Fig 5E) indicating that local histone arginine symmetric dimethylation catalyzed by PRMT5 at the *TCF3* splicing locus is sufficient to alter the alternative splicing pattern of *TCF3*. Put together, these results suggested that PRMT5-mediated histone symmetric dimethylation is an important and a necessary event in the regulation of *TCF3* splicing.

Next, we aimed to investigate the mechanism by which PRMT5-mediated histone methylation resulted in *TCF3* exon 18A exclusion under hypoxia. Interestingly, we identified a CpG island within the *TCF3*-ICR region (S4E Fig) which we hypothesized could facilitate DNA methylation and subsequently influence TCF3 splicing. Therefore, we first performed MeDIP-qPCR analysis in both MCF7 and MDA-MB-231 cells to understand the DNA methylation pattern in hypoxia at the *TCF3*-ICR region. We observed a significant increase in DNA methylation at the *TCF3*-ICR region under hypoxia (Figs 5F and S4F) indicating that PRMT5-mediated symmetric arginine dimethylation of histones was somehow

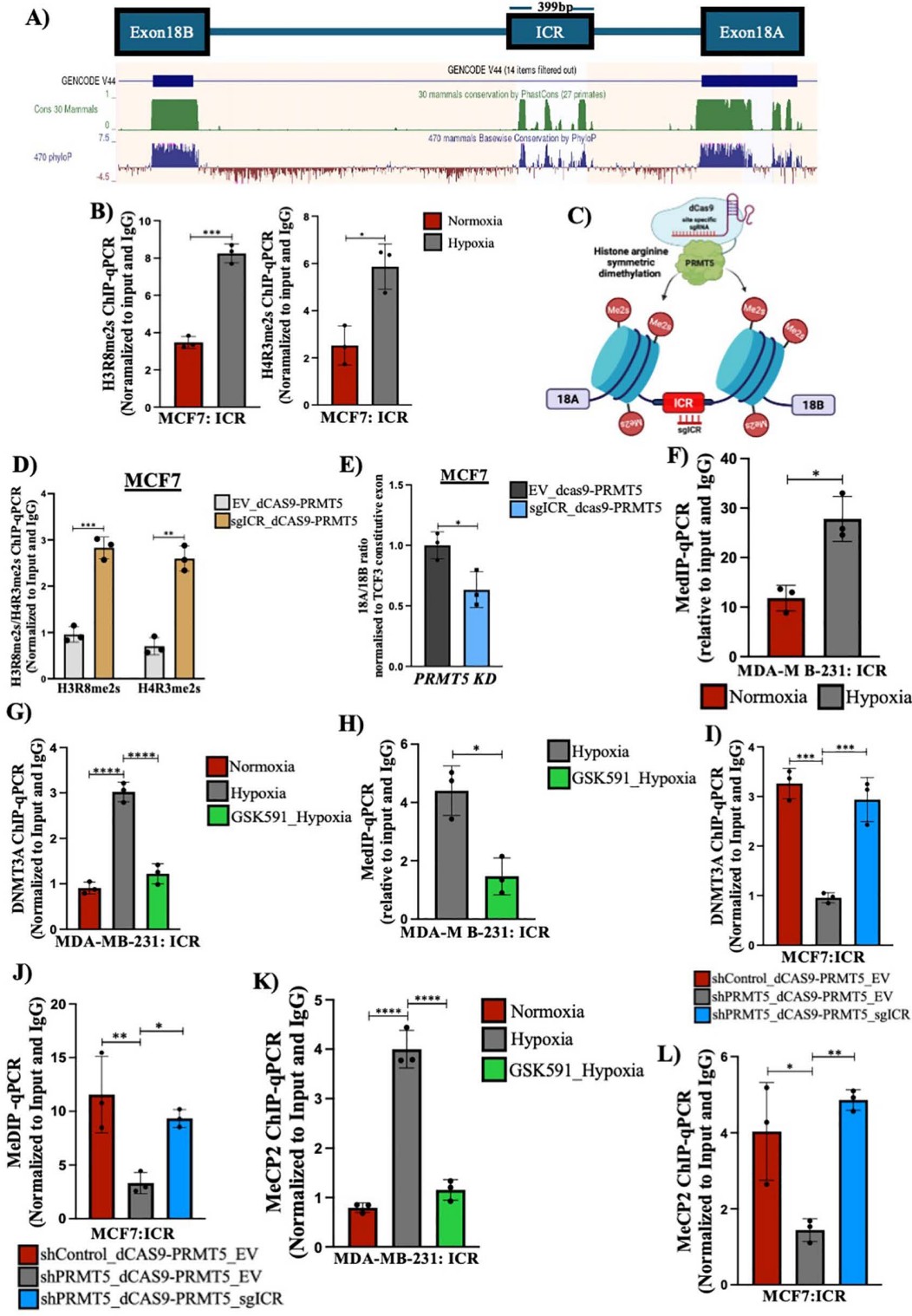

**Fig 5. PRMT5-mediated histone methylation regulates *TCF3* alternative splicing under hypoxia. A)** Genome track from UCSC genome browser showing TCF3 Intronic Conserved Region (ICR). **B)** H3R8me2s and H4R3me2s ChIP qPCR depicting increased histone symmetric arginine dimethylation at TCF3-ICR region in MCF7 cells, normoxia vs. hypoxia. **C)** Schematic representation of the dCAS9-PRMT5 epigenome editing vector targeted at the TCF3-ICR region. **D)** H3R8me2s and H4R3me2s ChIP qPCR depicting increased histone symmetric arginine dimethylation at TCF3-ICR region

after transfection with dCAS9-PRMT5-EV vs. dCAS9-PRMT5-sgICR in MCF7 shPRMT5 cells. **E)** qRT-PCR showing decrease in exon 18A/18B ratio after transfection with dCAS9-PRMT5-EV vs. dCAS9-PRMT5-sgICR in MCF7 shPRMT5 cells. **F)** MeDIP qPCR showing increase in DNA methylation at TCF3-ICR region in MDA-MB-231 cells, normoxia vs. hypoxia. **G)** DNMT3A ChIP qPCR showing change in DNMT3A binding at TCF3-ICR region in MDA-MB-231 cells treated with DMSO (Nx vs. Hx) or 5 μM GSK591(Hx). **H)** MeDIP qPCR showing decrease in DNA methylation at TCF3-ICR region in MDA-MB-231 cells after treatment with DMSO vs. 5 μM GSK591under hypoxia. **I)** DNMT3A ChIP qPCR showing change in DNMT3A binding at TCF3-ICR region after transfection with dCAS9-PRMT5 construct in PRMT5-depleted MCF7 cells under hypoxia. **J)** MeDIP qPCR showing increase in DNA methylation at TCF3-ICR region after transfection with dCAS9-PRMT5 construct in PRMT5-depleted MCF7 cells under hypoxia. **K)** MeCP2 ChIP qPCR depicting change in MeCP2 binding at TCF3-ICR region in MDA-MB-231 cells treated with DMSO (Nx vs. Hx) or 5 μM GSK591(Hx). **L)** MeCP2 ChIP qPCR depicting change in MeCP2 binding at TCF3-ICR region after transfection with dCAS9-PRMT5 construct in PRMT5-depleted MCF7 cells under hypoxia. Error bars, mean ± SEM; two-tailed $t$ test, one-way ANOVA. $*p < 0.05$, $**p < 0.01$, $***p < 0.001$, $***p < 0.0001$, $n = 3$ biological replicates. Numerical data of (B), (D–L) available in S1 Data, sheet "Figure 5." 5C, Created in BioRender. Shukla, S. (2025) https://BioRender.com/jcru48k.

responsible for subsequent DNA methylation at the TCF3-ICR region. Next, we wanted to understand how PRMT5-mediated histone methylation was responsible for DNA methylation at the ICR region. PRMT5-mediated H4R3me2s modification is known to recruit DNMT3A [30]. However, there were no reports suggesting the involvement of such mechanism in the regulation of alternative splicing. Since we observed an enrichment of H4R3me2s mark at the TCF3-ICR along with increased DNA methylation, we wanted to check if PRMT5-mediated histone arginine symmetric dimethylation of H4R3 (H4R3me2s) could recruit DNMT3A and subsequently methylate the DNA. To check this, we first performed DNMT3A ChIP-qPCR analysis in both MCF7 and MDA-MB-231 cells under hypoxia with or without GSK591 treatment to simultaneously establish the importance of PRMT5-mediated histone arginine symmetric dimethylation in recruitment of DNMT3A to the target locus. We found a significant enrichment of DNMT3A at the TCF3-ICR region under hypoxia which was significantly downregulated upon GSK591 treatment (Figs 5G and S4G). In correspondence with the decrease in DNMT3A recruitment upon GSK591 treatment, we also observed a significant downregulation in the DNA methylation at TCF3-ICR region upon ablation of PRMT5 methyltransferase activity (Figs 5H and S4H) indicating that under hypoxia, PRMT5-mediated H4R3me2s mark recruits DNMT3A to the TCF3 splicing locus and is responsible for DNA methylation. In addition, we made use of the dCAS-PRMT5 construct to analyze the importance of PRMT5-mediated histone methylation in recruiting DNMT3A followed by subsequent DNA methylation at the TCF3-ICR region. We depleted the endogenous PRMT5 under hypoxia in MCF7 cells and targeted the dCAS-PRMT5 to the ICR region and found that DNMT3A recruitment was rescued upon dCAS9-PRMT5 treatment (Fig 5I). Correspondingly, we also observed that the DNA methylation at the ICR region also was rescued (Fig 5J) indicating that the first step of PRMT5-mediated histone methylation is crucial for the recruitment of DNMT3A and subsequent DNA methylation at the TCF3-ICR region. Together, these results show that PRMT5 symmetrically dimethylates H4R3 under hypoxia which in turn recruits DNMT3A which further facilitates DNA methylation of the TCF3-ICR region. We also observed that abrogation of the histone mark (H4R3me2s) using GSK591 was sufficient to inhibit DNMT3A recruitment and DNA methylation at the TCF3-ICR region.

DNA methylation at the splicing locus attracts DNA-binding proteins resulting in altered RNA Pol II kinetics causing differential alternative splicing outcomes [15,46]. Methyl-CpG-Binding Protein 2 (MeCP2) preferentially binds to methylated DNA sequences and is known to dictate alternative splicing outcomes [56,57]. Since we observed an increase in DNA methylation at the TCF3-ICR region, we wanted to check if MeCP2 was able to bind at the TCF3-ICR region and regulate TCF3 alternative splicing event under hypoxia. Hence, we performed MeCP2 ChIP qPCR analysis with or without GSK591 treatment under hypoxia to check if there was any MeCP2 binding as a result of enriched DNA methylation at the TCF3-ICR. Certainly, we observed a significant enrichment of MeCP2 at the TCF3-ICR region under hypoxia which was abolished upon treatment with the PRMT5 inhibitor GSK591 (Figs 5K and S4I). Once again, we observed that targeting the ICR region under hypoxia in PRMT5-depleted MCF7 cells using the dCAS9-PRMT5 construct resulted in rescue of MeCP2 binding under hypoxia (Fig 5L). In order to further validate the role of MeCP2 in regulating TCF3 splicing, we performed shRNA-mediated MECP2 knockdown under hypoxia (S4J Fig) and observed changes in TCF3 splicing pattern. Our qRT-PCR results showed a significant increase in exon 18A/18B ratio upon MeCP2 downregulation under hypoxia

(S4K Fig) indicating that MeCP2 is essential for exon 18A exclusion under hypoxia. Collectively, these results indicated that PRMT5-mediated H4R3me2s modification at the *TCF3*-ICR region recruited DNMT3A which subsequently led to DNA methylation at the *TCF3*-ICR region under hypoxia. Binding of MeCP2 to the methylated DNA at the *TCF3*-ICR resulted in the exclusion of exon 18A under hypoxia.

### MeCP2 binding at the methylated *TCF3*-ICR region affects RNA Pol II kinetics under hypoxia

Since we demonstrated that MeCP2 binds to the methylated *TCF3*-ICR region, we next checked its role in altering RNA pol II kinetics, thereby affecting *TCF3* splicing under hypoxia. We performed RNA pol II ChIP-qPCR analysis with or without GSK591 treatment in both MCF7 and MDA-MB-231 cell lines under hypoxia to understand the change in RNA pol II occupancy levels at the *TCF3*-ICR region and its dependence on PRMT5-mediated histone arginine symmetric dimethylation. We performed RNA pol II ChIP qPCR and compared the enrichment of RNA pol II at the ICR with respect to sites upstream and downstream of the ICR region. This comparison is an indicative of the change in RNA pol II kinetics and suggests that a site showing more RNA pol II occupancy is likely because of RNA pol II pausing [15,46,58]. We observed a significant increase in RNA pol II occupancy at the *TCF3*-ICR region compared to sites upstream and downstream of the ICR region under hypoxia. The RNA pol II occupancy further decreased at the ICR region upon GSK591 treatment in both MCF7 and MDA-MB-231 cell lines (Figs 6A and S5A). These results suggested that binding of MeCP2 at the methylated *TCF3*-ICR region affected RNA pol II kinetics and that PRMT5-mediated histone methylation is critical in initiating the cascade of events which follow. Additionally, our dCAS9-PRMT5-mediated targeting of ICR region in PRMT5-depleted MCF7 cells under hypoxia showed an increase in RNA pol II occupancy at the *TCF3*-ICR region (Fig 6B) strengthening our observation that PRMT5-mediated histone methylation at the ICR region initiates a cascade of events which eventually regulate TCF3 splicing. Further, to verify that increase in RNA pol II occupancy at the *TCF3*-ICR is dependent on MeCP2, we performed shRNA-mediated knockdown of *MECP2* in MCF7 cells under hypoxia followed by RNA Pol II ChIP-qPCR. Our ChIP-qPCR analysis showed a significant reduction in RNA pol II occupancy at the *TCF3*-ICR region upon MeCP2 downregulation under hypoxia (Fig 6C) which suggested that RNA pol II elongation rate is affected by MeCP2 binding and could be a critical factor in orchestrating the *TCF3* splicing event.

In summary, we so far demonstrated that the ICR present between exons 18A and 18B serves as a site of control for *TCF3* splicing event. PRMT5-mediated H4R3me2s modification at the *TCF3*-ICR region increases under hypoxia which then recruits DNMT3A. Recruitment of DNMT3A subsequently results in increased DNA methylation at the *TCF3*-ICR region which serves as a binding site for MeCP2. Binding of MeCP2 to the methylated DNA results in slower RNA Pol II elongation rate which then becomes a critical factor in determining the *TCF3* splicing event.

### Increased RNA Pol II occupancy at the *TCF3*-ICR region recruits PTBP1 to affect *TCF3* splicing under hypoxia

Our observations so far suggested that under hypoxia, PRMT5-mediated symmetric arginine dimethylation of histones and subsequent DNA methylation, along with RNA pol II pausing at the *TCF3*-ICR region, led to the exclusion of the upstream 18A exon. Previously, it was shown that slowing of RNA Pol II could recruit a negative splicing factor, causing the exclusion of the upstream exon [59]. Therefore, we investigated whether RNA Pol II pausing at the *TCF3*-ICR region could possibly recruit a negative splicing factor, which could lead to the exclusion of exon 18A under hypoxia. To check this, we first performed a motif analysis of the *TCF3*-ICR region and found PTBP1 to have the highest binding score (S5B Fig). Therefore, we questioned whether PTBP1 could be the negative splicing factor responsible for exon 18A exclusion under hypoxia. To verify the involvement of PTBP1 in governing *TCF3* splicing, we performed shRNA-mediated knockdown of PTBP1 under hypoxia in both MCF7 and MDA-MB-231 cell lines (S5C Fig) and checked if it affected *TCF3* splicing. Our qRT-PCR analysis revealed a significant increase in the exon 18A/18B ratio upon PTBP1 knockdown under hypoxia (Fig 6D), indicating that the loss of PTBP1 resulted in increased exon 18A inclusion. This suggested that PTBP1

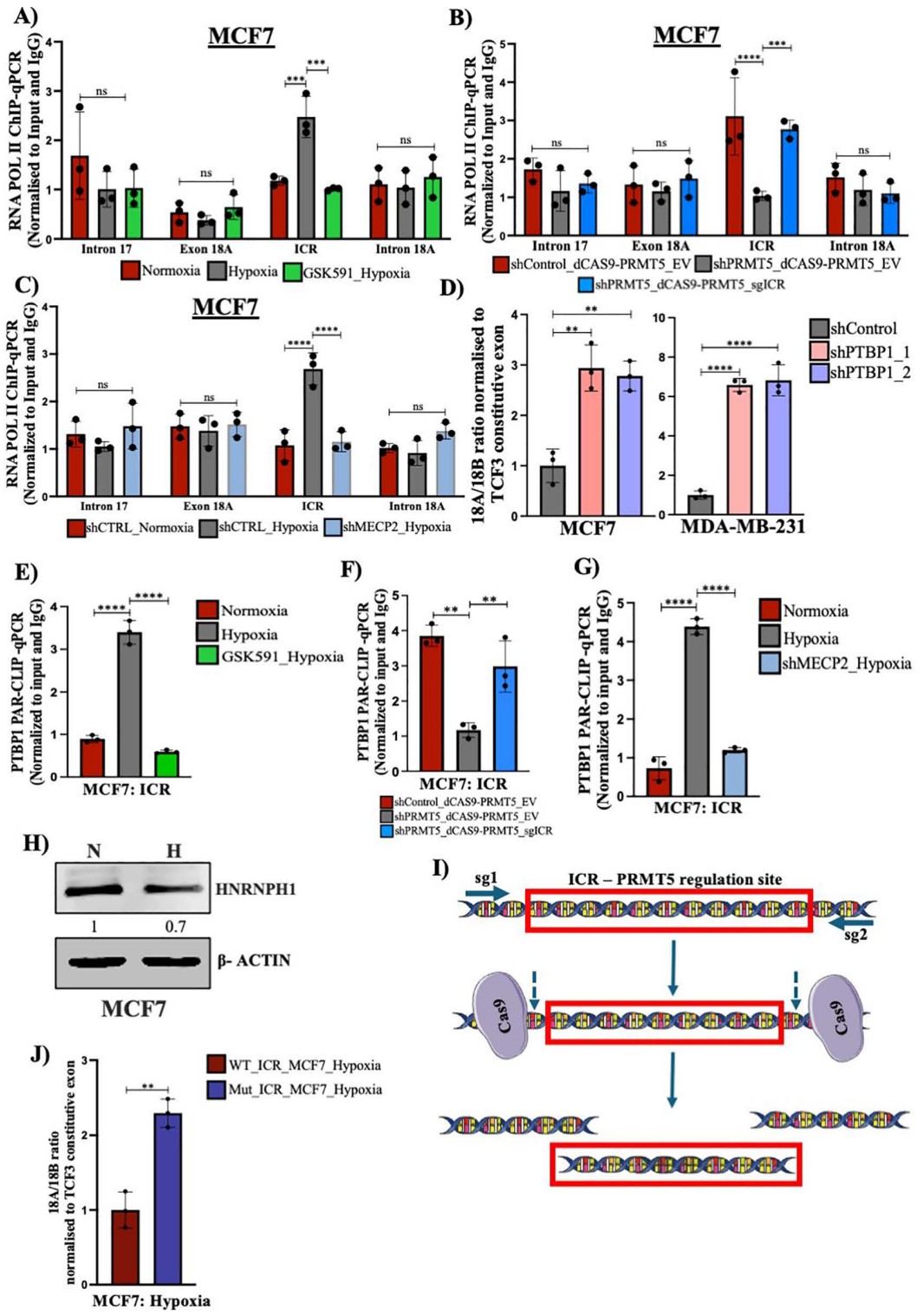

**Fig 6. Change in RNA pol II kinetics regulates *TCF3* splicing under hypoxia. A)** RNA pol II ChIP qPCR demonstrating change in RNA pol II occupancy at TCF3-ICR region in MCF7 cells treated with DMSO (Nx vs. Hx) or 5 μM GSK591(Hx). **B)** RNA pol II ChIP qPCR showing change in RNA pol II occupancy at TCF3-ICR region after transfection with dCAS9-PRMT5-EV vs. dCAS9-PRMT5-sgICR in PRMT5-depleted MCF7 cells. **C)** RNA pol II ChIP qPCR showing change in RNA pol II occupancy at TCF3-ICR region shCTRL (Nx vs. Hx) and shMECP2 (Hx) MCF7 cells. **D)** qRT-PCR showing increase

in exon 18A/18B ratio in shPTBP1, MCF7 and MDA-MB-231 cells under hypoxia. **E)** PTBP1 PAR-CLIP qPCR showing change in PTBP1 binding on RNA of *TCF3* at ICR region in MCF7 cells after treatment with DMSO vs. 5 μM GSK591under hypoxia. **F)** PTBP1 PAR-CLIP qPCR showing change in PTBP1 binding on RNA of *TCF3* at ICR region in after transfection with dCAS9-PRMT5-EV vs. dCAS9-PRMT5-sgICR in PRMT5-depleted MCF7. **G)** PTBP1 PAR-CLIP qPCR showing change in PTBP1 binding on RNA of *TCF3* at ICR region in shCTRL (Nx and Hx) vs. shMECP2 (Hx) MCF7 cells. **H)** Immunoblot depicting reduction in hnRNPH1 levels under hypoxia in MCF7 cells. **I)** Schematic showing the CRISPR/Cas9 strategy used for deletion of TCF3-ICR regulatory region which is important for PRMT5-mediated dictation of TCF3 alternative splicing under hypoxia. **J)** qRT-PCR showing increase in exon 18A/18B ratio in ICR regulatory region mutated MCF7 cells under hypoxia. Error bars, mean ± SEM; two-tailed *t* test, one-way ANOVA. *$p < 0.05$, **$p < 0.01$, ***$p < 0.001$, ****$p < 0.0001$, $n = 3$ biological replicates. Numerical data of **(A–G)** available in S1 Data, sheet "*Figure 6*".

is a critical regulator of *TCF3* splicing which mediates exon 18A exclusion under hypoxia. To establish that PTBP1 was indeed recruited to the *TCF3*-ICR region at the RNA, we performed PAR-CLIP-qPCR analysis under hypoxia with or without GSK591 treatment in MCF7 cell line. We observed a significant enrichment of PTBP1 binding on *TCF3* RNA at the *TCF3*-ICR region, which was decreased upon GSK591 treatment (Figs 6E and S5D), suggesting that PTBP1 recruitment was also dependent on PRMT5-mediated histone arginine symmetric dimethylation. Similarly, our dCAS9-PRMT5 targeted to the ICR region in PRMT5-depleted MCF7 cells under hypoxia rescued PTBP1 binding at the ICR region (Fig 6F) further establishing that PRMT5-mediated histone methylation is crucial for regulation of *TCF3* splicing under hypoxia. Having hypothesized that RNA Pol II pausing allows PTBP1 recruitment to *TCF3* premRNA, our next objective was to establish the significance of RNA Pol II pausing in PTBP1 recruitment. Since we had established that MeCP2 binding at the *TCF3*-ICR region resulted in increased RNA Pol II occupancy, we performed shRNA-mediated knockdown of *MECP2* in MCF7 cell line followed by PTBP1 PAR-CLIP-qPCR analysis to observe its subsequent effect on PTBP1 recruitment. Our PAR-CLIP-qPCR analysis revealed a significant decrease in PTBP1 enrichment on *TCF3* RNA at the *TCF3*-ICR region in sh*MECP2* MCF7 cells under hypoxia (Fig 6G) indicating that RNA Pol II pausing indeed regulates PTBP1 recruitment to the *TCF3*-ICR splicing locus under hypoxia. Put together, these observations demonstrated that MeCP2-mediated RNA pol II pausing was a critical step which facilitated PTBP1 recruitment to the *TCF3* splicing locus thereafter affecting *TCF3* splicing event under hypoxia. In addition to the role of PTBP1 in regulating TCF3 splicing, a previous report from [55] underscored the importance of change in hnRNPH1 levels also being a determining factor regulating exon 18A exclusion. Therefore, we analyzed the expression status of hnRNPH1 under hypoxia to understand how it affected *TCF3* exon 18A exclusion. Our immunoblotting analysis revealed a reduction in the protein levels of hnRNPH1 (Fig 6H). We speculate that reduced hnRNPH1 levels assist in exclusion of Exon18A under hypoxia. As mentioned in [55], reduced hnRNPH1 levels promote Exon18B inclusion which is consistent with our observations. Under hypoxia, PRMT5-mediated histone methylation subsequently recruits PTBP1 to the ICR region that eventually represses 18A inclusion under hypoxia. In normoxic condition or in the absence of PRMT5, PTBP1 fails to bind to ICR which allows hnRNPH1 to suppress 18B inclusion thereby promoting 18A inclusion.

Our study so far elucidated that the PRMT5-mediated symmetric arginine dimethylation of histones at the ICR, regulates TCF3 alternative splicing under hypoxia via inducing a cascade of events. Therefore, we further wanted to establish the role of this regulatory region of ICR which is important for PRMT5-mediated DNA methylation and subsequent MeCP2 binding. To achieve this, we intended to delete this ICR regulatory region endogenously to check its effect on the alternative splicing pattern of TCF3 under hypoxia. We made use of the CRISPR/Cas9 approach to excise the ICR regulatory region containing DNA methylation and MeCP2-binding sites (Fig 6I). We designed two sgRNAs flanking the ICR regulatory region (Fig 6I) which resulted in a successful deletion of our region of interest (S5E and S5F Fig). We then treated these ICR mutant cells under hypoxia and performed qRT-PCR analysis to check the effect of the deletion of the regulatory ICR region. We observed that the exon 18A/18B ratio was significantly upregulated in ICR mutant cells under hypoxia compared to wild type cells (Fig 6J) indicating that the regulatory region of ICR is important for PRMT5 to exert its effects to regulate of TCF3 alternative splicing under hypoxia. Together, these results highlighted that the regulatory region of the TCF3-ICR is indeed important for regulation of TCF3 alternative splicing under hypoxia.

## Alternative splicing of *TCF3* promotes EMT and invasion of breast cancer cells

Having established that PRMT5 regulates *TCF3* alternative splicing under hypoxia, we next wanted to study the importance of the *TCF3* alternative splicing event and its role in regulation of EMT and invasion under hypoxia. Under hypoxia, we observed that PRMT5 promoted exclusion of exon 18A and thereby leading to the production of the pro-invasive *TCF3*-18B (E47) isoform. *TCF3*-18B (E47) isoform is known to inhibit E-CAD expression promoting cell invasion [51,54,60,61]. However, there have been no reports of TCF3-mediated regulation of invasion under hypoxia. To address this, we first cloned both and *TCF3*-18A and *TCF3*-18B isoforms into pCMV-3tag-1A FLAG-tag vector (S5D Fig). Since normoxia mimics low *TCF3*-18B (E47) isoform condition, we overexpressed *TCF3*-18A (E12) and *TCF3*-18B (E47) isoforms independently in MCF7 cell line. We then performed immunoblotting analysis to observe the change in E-CAD protein expression. Expectantly, our immunoblotting analysis revealed a decrease in E-Cad protein levels upon *TCF3*-18B (E47) isoform overexpression (S6A Fig) indicating that TCF3-18B (E47) isoform induces EMT. Further to check the effect of E47 isoform on the invasion of breast cancer cells, we overexpressed FLAG-tagged *TCF3*-18A (E12) and *TCF3*-18B (E47) isoforms (Fig 6B and 6C) and performed Matrigel invasion assay in both MCF7 and MDA-MB231 cell lines. We observed that overexpression of *TCF3*-18B (E47) isoform led to a significant increase in invasive potential of both MCF7 and MDA-MB-231 cells indicating that *TCF3*-18B (E47) isoform of *TCF3* gene promotes tumor cell invasion (Fig 6D). Together, these results suggested that PRMT5-mediated regulation of *TCF3* alternative splicing under hypoxia, led to the production of E47 isoform which essentially promoted EMT and invasion under hypoxia. Since we had established that PRMT5-mediated H4R3me2s mark is sufficient to regulate *TCF3* splicing which subsequently regulates tumor cell invasion, we asked if targeting dCAS9-PRMT5 to the *TCF3* splicing locus will be sufficient to enhance invasion by altering *TCF3* splicing event. We performed Matrigel invasion assay, and we observed a significant increase in cell invasion upon transfection with dCAS9-PRMT5 (S6E Fig) under hypoxia indicating that PRMT5-mediated histone modification at the *TCF3*-ICR splicing locus is sufficient to induce cell invasion. This observation was indicative of the importance of histone methylation in regulating *TCF3* alternative splicing under hypoxia.

## Discussion

Epigenetic landscape alteration and aberrant alternative splicing aid in regulation of important tumorigenic events like EMT and invasion [62–67]. Epigenetic changes offer flexibility and heritability to cancer cells to weather the mounting demands of a repugnant tumor microenvironmental condition such as hypoxia [68–70]. Also, alternative splicing offers transcript diversity and novelty in dealing with harsh conditions like hypoxia [3,6,7,12,71]. Since alternative splicing event is cotranscriptional, several studies have explored the role of epigenetic features at the local transcriptional locus in dictating alternative splicing outcomes [14–16,59]. However, the role of microenvironmental stress conditions like hypoxia in regulating epigenetics-mediated changes in alternative splicing has been significantly less explored.

PRMT5 is known to aid tumor progression [28,72–75], yet its role and regulation in hypoxia remains underexplored. While PRMT5 is known to regulate gene expression and alternative splicing via its ability to methylate histones [20,30] and spliceosome complex proteins independently [27,74,76] the ability of PRMT5 in regulating alternative splicing via modulation of histone methylation under hypoxia is not yet studied. Therefore, our research aimed at elucidating the mechanisms involved in regulation of PRMT5 expression under hypoxia and hypoxia-induced regulation of alternative splicing via PRMT5-mediated histone methylation.

In this study, we initially observed that among all PRMTs, PRMT5 was upregulated in breast cancer patients as well as in breast cancer cell lines. Further, we systematically deduced that CCCTC-binding factor (CTCF) binds at the demethylated *PRMT5* promoter under hypoxia and mediates transcriptional upregulation of *PRMT5*. PRMT5 upregulation concordantly resulted in an increase in PRMT5-mediated symmetric arginine dimethylation of histones H3R8 and H4R3 (H3R8me2s/H4R3me2s). These observations highlighted that PRMT5 is upregulated under hypoxia which can have functional consequences in tumor progression. We then assessed the role of PRMT5 in regulating EMT and invasion

under hypoxia. We observed a PRMT5-dependent induction of EMT and tumor cell invasion under hypoxia. Subsequently, we demonstrated that PRMT5-mediated symmetric arginine dimethylation is important in regulation of EMT induction and invasion under hypoxia as inhibition of PRMT5 methyltransferase activity using GSK591 was sufficient to curb EMT and invasion under hypoxia. Our study portrays a comprehensive picture of the mechanism of PRMT5 upregulation under hypoxia, and its importance in promoting tumorigenic events such as EMT and invasion. Having established that PRMT5-mediated histone methylation is crucial in promoting EMT under hypoxia, our aim was to investigate the influence of epigenetic modifications in dictating alternative splicing outcomes under hypoxia. To study this, we performed RNA-Seq analysis upon PRMT5 downregulation and found that under hypoxia, PRMT5 led to significant changes in the alternative splicing patterns. Our gene ontology analysis of the genes undergoing PRMT5-dependent mutually exclusive exon (MXE) splicing revealed regulation of apical junction and EMT as one of the top processes regulated by PRMT5 further indicating that PRMT5-regulated EMT and invasion under hypoxia via regulation of alternative splicing. Among all the transcription factors responsible for inducing EMT, TCF3, a bHLH domain-containing transcription factor was identified as one of the novel targets of PRMT5-mediated alternative splicing. The 18A and 18B exons of *TCF3* undergo alternative splicing to give rise to either *TCF3*-18A (E12) isoform or *TCF3*-18B isoform (E47). *TCF3*-18B isoform inhibits E-CAD expression thereby promoting EMT and invasion. We observed a significant decrease in exon 18A/18B ratio upon ablation of PRMT5 expression. Also, breast cancer cells treated with GSK591 showed decreased TCF3 exon 18A/18B ratio under hypoxia which indicated that PRMT5-mediated symmetric dimethylation is important for regulation of *TCF3* alternative splicing event. Our further analysis revealed the presence of an ICR [55] present between exon 18A and exon 18B of *TCF3* gene which was a potential site of regulation. At the intronic conserved site, we observed an increase in PRMT5-mediated symmetric arginine dimethylation marks H3R8me2s and H4R3me2s under hypoxia. Using the dCAS9-PRMT5 epigenome editing system, we showed that these marks were necessary and sufficient for TCF3 alternative splicing regulation under hypoxia. PRMT5-mediated increase in H4R3me2s mark recruited DNMT3A followed by subsequent DNA methylation at the ICR under hypoxia. Methylated DNA was recognized and bound by MeCP2 which resulted in RNA pol II pausing and subsequent recruitment of the negative splicing factor PTBP1 onto the RNA at the *TCF3*-ICR. Under hypoxia, PTBP1 led to the exclusion of exon 18A leading to the production of the pro-invasive TCF3-18B (E47) isoform. In addition to the complex, epigenetics-mediated regulation of alternative splicing of *TCF3*, we believe that the reduction in hnRNPH1 levels under hypoxia is also an important aspect which plays a crucial and decisive role in determining the splicing outcome of *TCF3* exon 18A/18B. This further emphasizes the complex multilayered nature of splicing regulation. We believe that such complexity involving epigenetic modulation and expression of splicing factors can act as molecular switches to dynamically adjust to stress conditions like hypoxia.

As reported by different studies, the TCF3/E2A gene undergoes an alternative splicing event to produce two isoforms E12 and E47 [51]. Both these isoforms are class I bHLH transcription factors which bind to E box regions and regulate transcription. As a result of the alternative splicing, the two isoforms harbor a slightly different bHLH DNA-binding domain in the protein sequence [72]. This apparent difference in the bHLH domain can vary the degree of DNA binding ability of both these isoforms [77]. Previous reports have suggested that E47 has a greater binding affinity to DNA compared to the E12 isoform [77]. The E12 isoform has been reported to be indispensable during development whereas E47 isoform has specific roles in development [78,79]. In the context of breast cancer, the E47 isoform has been reported to promote stemness and tumor invasion [54,80]. Taken together, we speculate that E12 and E47 might bind differently to DNA in normoxic condition compared to hypoxia, respectively. It would be interesting to evaluate if epigenetic modifications have any role in determining the binding specificities of these proteins.

In summary, we demonstrated that PRMT5 is upregulated and promoted EMT and invasion under hypoxia in breast cancer. PRMT5-mediated symmetric arginine dimethylation of H4R3 (H4R3me2s) at the *TCF3* splicing locus is responsible for DNMT3A-MECP2-PTBP1-mediated regulation of TCF3 alternative splicing leading to the production of a pro-invasive TCF3-18B isoform which thereby promotes EMT in breast cancer hypoxia (Fig 7). Our study provides a unique

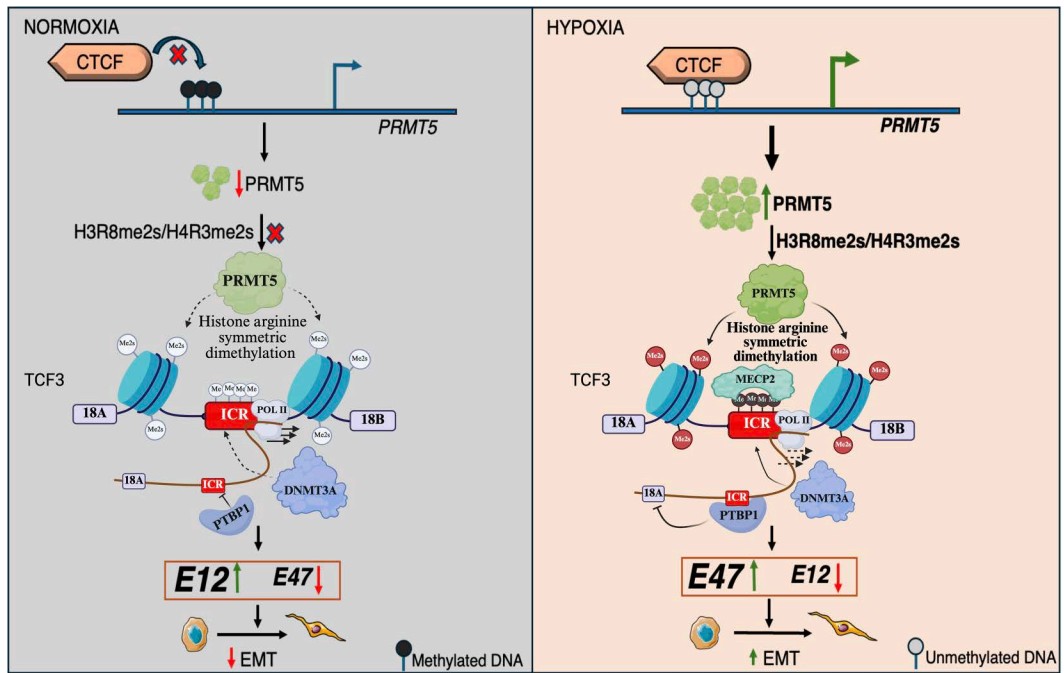

**Fig 7. Schematic model depicting the role of PRMT5-mediated symmetric arginine dimethylation in regulation of TCF3 alternative splicing under hypoxia.** *Created in BioRender. Shukla, S. (2026) https://BioRender.com/7hvj1qz* https://BioRender.com/2lkb2sk.

outlook on the triangulation of hypoxia, epigenetics, and alternative splicing aiding tumor progression. Also, our mechanistic insights into *TCF3* splicing reveal the intimate connection between histone marks, DNA methylation, and splicing factor recruitment at the splicing locus throwing light on the innate complexity of mRNA splicing. Overall, our results highlight the role of PRMT5 in orchestrating cellular adaptation under hypoxia by regulating alternative splicing via histone methylation. This study opens new avenues for understanding hypoxia-driven tumor progression through the interplay of PRMT5-mediated histone and DNA methylation, as well as alternative splicing. These regulatory networks could be potential targets for development of innovative therapeutic strategies against hypoxia-inflicted breast tumors.

## Materials and methods

### Ethics statement

All the breast cancer patient data and tissue sections collected for this study was approved by the Institute ethics committee of Indian Institute of Science Education and Research-Bhopal, India (IISERB/IEC/Certificate/2021-I/09) dated December 10th, 2021. The written informed consents were taken from all the patients before collecting tissue samples.

### Cell culture

Human breast cancer cell lines MCF7 and MDA-MB-231 were obtained from American Type Culture Collection (ATCC). BC8322 primary cell line derived from breast cancer patient as described in [5]) was used in this study. MCF7, MDA-MB-231, BC8322 and HEK293T were cultured at 37°C, 5% $CO_2$ in DMEM (Invitrogen, 12,800,017, lot no. 2248833) supplemented with 10% fetal bovine serum (FBS; Sigma, F7524, lot no. BCBX8466), 100 units/ml of penicillin and streptomycin (Invitrogen, 15,140,122, lot no. 2321120), and 2 mM/l L-glutamine (Invitrogen, 25,030,081, lot no. 1917006). The cell lines used in the study were routinely tested for mycoplasma contamination using PCR based method. For GSK591

(Cayman Chemical, 18,354) treatment, 5 µM GSK591dissolved in DMSO was added to media used for culturing cells. Cells were incubated with GSK591 for a total of 24 hours. Cells were maintained at 1% $O_2$ for hypoxia treatment in a Ruskinn INVIVO2 hypoxia chamber.

### Plasmids

The PRMT5 overexpression construct was generated by cloning the insert between *BamHI* and *EcoRI* sites of the plasmid pCMV-3tag-1a (Agilent, 240195). TCF3-18A and TCF3-18B overexpression constructs were cloned into pCMV-3tag-1a using *BamHI* and *HindIII.* For creating the dCAS9-PRMT5 construct, firstly, the DNMT3A CD was digested out of the dCAS9-DNMT3A-puroV2 plasmid (Addgene, 74407) using *BamHI* and *FseI*. Following this, MCS taken from pCMV-3tag1A was annealed and cloned in place of DNMT3A CD to create a base parent vector named as dCAS9-MCS. Subsequently, PRMT5 full-length construct was cloned into the dCAS9-MCS using *BamHI* and *Sal1* to generate the dCAS9-PRMT5 construct. All of the constructs generated have been sequence verified using Sanger sequencing. Primers used in construction of these vectors have been listed in the supplementary S1 Table.

### dCAS9-mediated epigenome editing

Guide RNAs (sgRNA) were annealed and cloned at the BbsI restriction site downstream of the U6 promoter in dCAS9-DNMT3A-PuroR_v2 (Addgene, 74,407) [4,81] or dCAS9-PRMT5 vector (This paper). Nontargeting sgRNA containing dCAS9-DNMT3A vector (Addgene, 71830) was used along with dCAS9-DNMT3A system as control. Transfections were performed in 60 or 100 mm dishes using TurboFect (Thermo Scientific, R0531). For dCAS9-DNMT3A-mediated epigenetic editing studies, cells were subjected to hypoxia, 12 h post transfection. Following hypoxia treatment, cells were harvested for either immunoblot assay, MeDIP or ChIP analysis. For dCAS9-PRMT5-mediated epigenetic editing, cells were harvested 24 h post transfection for either ChIP qPCR analysis or RNA was collected for q-RT PCR analysis.

### Quantitative RT-PCR

Total RNA was extracted using TRIZol reagent (Ambion, 15,596,018, lot no. 260712) as per the manufacturer's instructions. cDNA was synthesized using PrimeScript 1st strand cDNA Synthesis Kit (Takara, 6110A). qRT-PCR reactions were setup using KAPA SYBR FAST QPCR master mix (Sigma, KK4618) in a qTOWER$^3$-G (Analytik Jena) qPCR machine according to the manufacturer's protocol. The primers were designed using the IDT Primer Quest tool (https://www.idtdna.com) and are enlisted in S1 Table. Housekeeping control gene *RPS16* was used to normalize using the formula $2^{(Ct\ Control\ -\ Ct\ Target)}$ [82] Student *t* test or one-way ANOVA was used to compare gene/exon expression between two and three different groups, respectively. $p < 0.05$ was considered statistically significant.

### Immunoblotting

Urea lysis buffer (8 M urea, 2 M thiourea, 2% CHAPS, 1% DTT) supplemented with 1 × protease inhibitor cocktail (PIC; leupeptin 10–100 µM, pepstatin 1 µM, EDTA 1–10 mM, AEBSF < 1 mM) and 10 mM PMSF was used to lyse the cells. Equal concentration of protein samples was loaded for every experiment. Recommended dilutions of primary antibodies were used to incubate blots overnight at 4°C, followed by 1 hour incubation with secondary antibody. The blots were scanned using Odyssey membrane Scanning system. Quantification of the bands was performed using ImageJ. The uncropped images of all the blots are available in S1 Raw Images file. Antibodies used for immunoblotting have been listed in S4 Table.

### PAR-CLIP-qPCR

PAR-CLIP was performed as described earlier [83] with slight modifications. 100 µM 4SU (4-thiouridine, Cayman Chemical, T4509) was added in the culture medium 14 h. prior to crosslinking. Cells were crosslinked at 0.15 J cm$^{-2}$ total

energy of 365-nm UV light using a UV-crosslinker. Cells were scraped in 3 mL phosphate-buffered saline (PBS) and were pelleted by centrifugation at 500$g$ for 5 min at 4°C. Following this, cells were resuspended in NP40 lysis buffer (50 mM HEPES-KOH, pH 7.5, 150 mM KCl, 2 mM EDTA-NaOH, pH 8, 1 mM NaF, 0.5% v/v NP40) supplemented with 1× PIC, 10 mM PMSF, 0.5 mM DTT, and 40 U/mL RNaseOUT (Invitrogen, 10777019). Cells were allowed to lyse in ice for 10 min and cleared at 13,000$g$ for 15 min at 4°C. Lysate was incubated with PTBP1, or rabbit IgG antibodies conjugated to Dynabeads Protein G (Invitrogen, Cat No. 10004D) overnight at 4°C. Following antibody incubation, immunoprecipitate complex was washed with IP wash buffer (50 mM HEPES-KOH, pH 7.5, 300 mM KCl, 0.05% v/v NP40, 0.5 mM DTT (fresh), 1× PIC, 10 mM PMSF, 40 U/mL RNaseOUT) and high salt wash buffer (50 mM HEPES-KOH, pH 7.5, 500 mM KCl, 0.05% v/v NP40, 0.5 mM DTT (fresh)), 1× PIC, 10 mM PMSF, 40 U/mL RNaseOUT). 20 µL aliquots of the immunoprecipitate complex were taken for immunoblot analysis. The remaining volume was subjected to decrosslinking in Proteinase K buffer (100 mM Tris–HCl, pH 7.5, 12.5 mM EDTA-NaOH, pH 8, 150 mM NaCl, 2% v/v SDS, 1.2 mg mL$^{-1}$ Proteinase K) at 55°C for 30 min. TRIZol was added to the mixture and RNA was isolated as per the manufacturer's instructions including 1 µL of Glycoblue Coprecipitant (Invitrogen, AM9515). RNA obtained was converted to cDNA with random hexamers using PrimeScript 1st strand cDNA Synthesis Kit (Takara, 6110A). qRT-PCR was performed using KAPA SYBR FAST QPCR master mix (Roche, KK4618). The enrichment of IP was normalized to its respective input using ($2^{(Ct\ IP\ -\ Ct\ Input)}$). Fold change was calculated relative to IgG enrichment for representation.

## Chromatin immunoprecipitation (ChIP) assay

ChIP assay was performed as previously described [5]. Briefly, about 10 million cells were crosslinked using 1% formaldehyde and scraped in PBS, followed by lysis and sonication. About 10–25 µg of sheared chromatin was incubated with an antibody of interest overnight at 4°C. Immunoprecipitation was performed using Dynabeads Protein G (Invitrogen, Cat No. 10004D). The immunoprecipitated protein–DNA complexes and 5% input were analyzed by qRT-PCR using KAPA SYBR FAST master mix (Sigma, KK4618). Normalization was performed using the formula [$2^{(Ct\ input\ -\ Ct\ immunoprecipitation)}$]. The obtained values were further normalized to the relative rabbit IgG and control IP values. Significance between the two groups was calculated using Student $t$ test or one-way ANOVA with a $p$ value of <0.05 considered statistically significant. Primers used for ChIP-qPCR analysis are listed in S1 Table. Antibodies used for ChIP assay have been listed in S4 Table.

## Immunohistochemistry-fluorescence (IHC-F)

This study was approved by the Institute Ethics Committee of Indian Institute of Science Education and Research-Bhopal, India (IISERB/IEC/Certificate/2021-I/09) dated December 10th, 2021. The tissue sections were obtained following the written consent from the patient. Breast tumor tissues of 5 microns size were obtained from Bansal Hospital, Bhopal, India, for immunohistochemical analysis. Briefly, the sections were deparaffinized at 65°C followed by xylene treatment and were rehydrated using ethanol gradients. Heat-induced antigen retrieval was performed in citrate buffer (pH 6.0) for 10 min. Blocking was done using 5% BSA for 1 h at RT. Primary antibodies against PRMT5 and CA9 were incubated overnight at 4°C. Subsequently, tumor sections were incubated with Alexa-Flour 555 anti-rabbit IgG secondary antibody for 1 h following which nuclear counter stain was performed using DAPI (Invitrogen, D1306, lot no. 1673432). Imaging was performed using Olympus FV3000 confocal laser scanning microscope with a 40× objective. Image analysis was performed using ImageJ software. The details of patient samples are listed in S3 Table.

## Luciferase reporter assay

PRMT5 promoter fragments 500, 1,000, 1,500, and 2,000 bp upstream of TSS were amplified individually. Genomic DNA (gDNA) obtained from MCF7 served as a template. Amplified fragments were then cloned between *KpnI* and *HindIII* restriction sites of the pGL3 basic expression vector (Promega, E1751). MCF7 or MDAMB231 cells were seeded in 24-well plates and were co-transfected with different PRMT5 promoter luciferase constructs along with pRL-TK renilla

luciferase plasmid (Promega, E2231) using PEI (Polyethyleneimine, 23966-100, Polysciences). Sixteen hours posttransfection, the cells were subjected to appropriate hypoxia treatment. Subsequently, the cells were lysed, and the luciferase activity was determined using GloMax- Multi Detection System (Promega) by normalizing to the Renilla luciferase activity. Primers used for amplification of different fragments are listed in S1 Table.

## Site-directed mutagenesis

To generate CTCF-binding site mutant in the PRMT5 promoter luciferase construct, SDM primers containing mutations as mentioned in S1 Table were used. PRMT5-1500 promoter construct was used as a template. Following *DpnI* digestion and transformation, positive colonies were screened and the presence of intended mutations in the CTCF-binding site were confirmed by Sanger sequencing.

## Matrigel invasion assay

Transwell with 8.0 μm pore Polycarbonate membrane insert was first coated with Matrigel (Corning, 356,230, lot no. 2010001). MCF7 or MDA-MB-231 cells ($3 \times 10^4$) were resuspended in serum-free media and were added onto the Matrigel layer of the insert. The insert was placed in a well containing media supplemented with 10% FBS. The whole setup was placed in a hypoxia incubator for 24–72 h. Following hypoxia incubation, cells were fixed with 4% formaldehyde and stained with crystal violet (0.05% in 10% methanol in 1× PBS). Images were taken at 20× in an inverted microscope (Olympus CKX41) and image analysis was performed using Image J. Numerical data of all replicates including the type of statistical analysis used and *p* values of all invasion assay analyses has been included in S1 Data.

## RNA interference

The lentivirus containing small hairpin RNA (shRNA) (Sigma, Mission Human Genome shRNA Library) against target genes along with the packaging plasmids were transfected into HEK293T cells using PEI. Lentivirus was collected 24 and 48 hours posttransfection. MCF7 and MDA-MB-23 cells were seeded at a density of $3 \times 10^5$ cells per well of a six-well culture plate. Lentivirus-containing media obtained from HEK293T cells was used to inoculate target cells along with 8 μg/ml polybrene (Sigma, H9268, lot no. SLBH5907V). Twelve hours postviral inoculation, the media was changed, and the cells were selected with 1 μg/ml puromycin (Sigma, P9620, lot no. 034M4008V) for 72 h following which cells were used for different assays as desired. The list of shRNA sequences used in this study are provided in S2 Table.

## Methylation-Dependent Immunoprecipitation (MeDIP)

gDNA was isolated using gDNA isolation kit (Sigma, G1N70) as per manufacturer's protocol. MeDIP was performed as described earlier [31]. About 3 μg of gDNA was sonicated using the Bioruptor sonicator (Diagenode) to obtain a fragment size approximately ranging from 100 to 400 bp. Subsequently, sonicated DNA was denatured at 95°C for 10 min and then incubated with 5methyl cytosine antibody and rabbit IgG antibody overnight at 4°C. Five percent of denatured DNA was taken as input. Immunoprecipitation was performed using Dynabeads Protein G (Invitrogen, 10004D). Decrosslinking of the immunoprecipitated complexes was performed in TE buffer with 1% SDS and Proteinase K (Invitrogen, 25,530,049) overnight at 65°C. De-crosslinked DNA and input was purified using a PCR purification kit (QIAGEN, 28,106). The immunoprecipitated samples along with input were analyzed by q-RT PCR using KAPA SYBR FAST (Sigma, KK4618). The enrichment was calculated using input as normalization factor and the following formula: $2^{(Ct\ input\ -\ Ct\ Immunoprecipitation)}$. Additionally, the resultant values were normalized by the enrichment values of normal rabbit IgG. The final values are represented as mean ± SD of triplicates. The details of primers used for MeDIP qPCR are provided in S1 Table.

## CRISPR/Cas9-mediated ICR regulatory region editing

To delete the ICR regulatory region necessary for PRMT5-mediated regulation of TCF3 alternative splicing, we made use of the CRISPR/Cas9 approach as reported in [84] with slight modifications. sgRNA were annealed and cloned at the *BsmbI* restriction site downstream of the U6 promoter in pLentiCRISPR-E (Addgene, 78852) plasmid. Two sgRNAs, one upstream of the ICR regulatory site (sg1) and one downstream of the regulatory site (sg2) were picked. HEK293T cells were transfected with equal concentration of plasmids containing sgRNA1 and sgRNA2 along with the packaging plasmids using PEI. Supernatant, containing viral particles was collected 48 h post transfection. Lentivirus-containing media obtained from HEK293T cells was used to inoculate MCF7 cells along with 8 µg/ml polybrene (Sigma, H9268, lot no. SLBH5907V). Twenty-four hours post viral inoculation, the media was changed, and the cells were selected with 1 µg/ml puromycin (Sigma, P9620, lot no. 034M4008V) for 72 h. After 72 h, gDNA was isolated and amplified using flanking primers. Amplified product was sent for sanger sequencing for Indel analysis. Indel analysis was performed using DECODR. Cells showing positive edits were subjected to hypoxia treatment and were further used for TCF3 splicing analysis. Sequences of sgRNAs and the flanking primers used for amplifying the edit region are provided in S1 Table.

## RNA-seq and bioinformatic analysis

Total RNA from hypoxia-treated sh*PRMT5* and shCTRL cells was isolated using PureLink RNA Mini Kit (Invitrogen, 12,183,025). One microgram of total RNA was used to make cDNA libraries. Prior to library preparation, mRNA enrichment was performed using NEBNext Poly(A) mRNA Magnetic Isolation Module (E7490S). The libraries were constructed using NEBNext Ultra II RNA Library Prep Kit for Illumina (E7770S).

Paired-end (2 × 150 bp) sequencing was performed on Illumina NovaSeqX platform (MedGenome Labs Ltd, Bengaluru, India). Raw sequencing reads were aligned to the human reference genome (hg38 assembly) with STAR Aligner (version 2.7.1a) [85]. For alternative splicing analysis, aligned reads were analyzed using rMATS 4.0.2 [86]. Differential splicing analysis was performed by calculating PSI ($|\Delta PSI| = |PSI\ (shCTRL \times 100) - PSI\ (shPRMT5 \times 100)| > 3\%$, $p < 0.05$) as described previously [W. juan Li and colleagues, 2023]. Genes undergoing mutually exclusive exon events upon PRMT5 knockdown under hypoxia have been listed in S1 File. Gene ontology was performed using ShinyGo 0.80 [87]. The list of genes in highlighted categories in listed in S1 File. List of EMT transcription factors was obtained from [50]. The list of EMT transcription factors has been provided in S1 File. Spearman's correlation analysis of *PRMT5* mRNA expression with hypoxia marker genes was performed using Gene Expression Profiling Interactive Analysis 2 (GEPIA 2) using TCGA BRCA tumor dataset [88] (http://gepia2.cancer-pku.cn/#correlation). Microarray data from GSE76250 was analyzed as described earlier [7,89,90]. The breast cancer patient microarray data from GSE76250 was downloaded from https://xenabrowser.net/. These samples were then stratified as either hypoxia high or hypoxia low based on signature gene expression obtained from Hypoxia Hallmark signature from MSigDB. Samples with higher scores (Top 20%) were stratified as hypoxia high whereas samples with lower scores (Bottom 20%) were stratified as hypoxia low following which gene expression levels for PRMTs was checked. Transcription factor binding sites with stringency set at 500 of breast cells were obtained from ChIP Atlas [91] (https://chip-atlas.org). UCSC genome browser hg38 tracks were used for CpG island and conservation analysis.

## ChIP-seq analysis

MCF7 cells CTCF ChIP-seq in normoxic versus hypoxic conditions was obtained from GSE216843 [4]. Raw fastq files were trimmed by Trimmoatic (v0.39) [92] with default parameters. Reads were uniquely aligned to GRCh38 using STAR (v2.7.3a) aligner [85]. The biological replicates were merged by Samtools v1.9 [93]. Peak calling was performed using MACS v2.1.2 with default parameters [94]. Significant CTCF binding peaks were scanned at PRMT5 promoter. Integrative Genomics Viewer (IGV) was used to visualize CTCF-binding peaks on PRMT5 promoter.

## Statistical analysis

Data are presented as mean ± SEM unless otherwise stated. At least three independent biological replicates have been performed for each experiment. All statistical analyses were conducted using GraphPad Prism 10 software. Statistical tests used and specific *p* values are indicated in the figure legends.

## Supporting information

**S1 Table. List of primers.**
(DOCX)

**S2 Table. List of shRNA sequence.**
(DOCX)

**S3 Table. Clinical information of patient samples.**
(DOCX)

**S4 Table. List of antibodies.**
(DOCX)

**S1 Raw Images. Raw images of Figs 1–6 and S2–S6.**
(PDF)

**S1 Data. Numerical values of data in Figs 1–6 and S2–S6 .**
(XLSX)

**S1 Fig. PRMT5 is upregulated under hypoxia in clinical breast cancer patient samples. A)** Immunohistochemistry-Fluorescence (IHC-F) analysis showing PRMT5 and CA9 expression in tumor sections of breast cancer patients. Scale bar 20 μm.
(DOCX)

**S2 Fig. CTCF is responsible for upregulation of PRMT5 under hypoxia in breast cancer cells. A)** Luciferase assay showing an increase in luciferase activity under hypoxia in MCF7 cells. **B)** Immunoblot showing decrease in PRMT5 expression upon CTCF KD under hypoxia in MDA-MB-231 cells. **C)** Transcription factors binding sites obtained from ChIP-Atlas showing presence of CTCF binding site at PRMT5 promoter in breast cells. **D)** MeDIP-qPCR showing decrease in DNA methylation at PRMT5 promoter in MDA-MB-231 cells normoxia versus hypoxia. **E)** CTCF Chip qPCR showing enrichment in CTCF binding at PRMT5 promoter in MDA-MB-231 cells normoxia versus hypoxia. **F)** Chromatogram showing mutations induced in the PRMT5 promoter luciferase (−1,500) construct. **G)** Luciferase assay showing decrease in luciferase activity in PRMT5 luciferase promoter construct (−1,500) harboring a mutated CTCF binding site. Error bars, mean ± SEM; two-tailed *t* test, one-way ANOVA. *$p < 0.05$, **$p < 0.01$, ***$p < 0.001$, ****$p < 0.0001$, $n = 3$ biological replicates. Numerical data of (A), (D–E), (G) available in S1 Data, sheet *"Figure S2."*
(DOCX)

**S3 Fig. PRMT5 regulates EMT and invasion under hypoxia. A)** qRT-PCR depicting mRNA expression of EMT markers in shCTRL versus shPRMT5 MDA-MB-231 cells under hypoxia. **B)** Immunoblot showing protein level of EMT markers in shCTRL versus shPRMT5 MDA-MB-231 cells under hypoxia. **C)** Immunoblot showing decrease in histone marks HR38me2s and H4R3me2s upon vehicle (DMSO) versus 5 μM GSK591 treatment in MDA-MB-231 cells under hypoxia. **D)** qRT-PCR depicting mRNA expression of EMT markers upon vehicle (DMSO) versus 5 μM GSK591 treatment in MDA-MB-231 cells under hypoxia. **E)** Immunoblot showing protein level of EMT markers upon vehicle (DMSO) versus 5 μM GSK591 treatment in MDA-MB-231 cells under hypoxia. **F, G)** Matrigel invasion assay and its respective quantification

in shControl versus shPRMT5 MDA-MB-231 cells under hypoxia. **H)** Matrigel invasion assay and its respective quantification (right side) upon vehicle (DMSO) versus 5 μM GSK591 treatment in MDA-MB-231 cells under hypoxia. Scale bar 2 μm. Error bars, mean ± SEM; two-tailed *t* test, one-way ANOVA. *$p < 0.05$, **$p < 0.01$, ***$p < 0.001$, ****$p < 0.0001$, $n = 3$ biological replicates. Numerical data of (A), (D) (G–H) available in S1 Data, sheet *"Figure S3."*
(DOCX)

**S4 Fig. PRMT5-mediated histone methylation regulates *TCF3* alternative splicing under hypoxia. A)** qRT-PCR showing change in exon 18A/18B ratio upon vehicle (DMSO) versus 5 μM GSK591 treatment in MDA-MB-231 cells. **B)** Immunoblot showing overexpression of PRMT5 in MCF7 cells under hypoxia. **C)** Vector map of dCAS9-PRMT5. **D)** Immunoblot showing Cas9 expression in MCF7 cells, untransfected control versus transfected with dCAS9-MCS or dCAS9-PRMT5 construct. **E)** H3R8me2s and H4R3me2s ChIP qPCR depicting increased histone symmetric arginine dimethylation at TCF3-ICR region in MDA-MB-231 cells, normoxia versus hypoxia. **F)** Genome track from UCSC genome browser showing the presence of a CpG island at the TCF3-ICR region. **G)** MeDIP qPCR showing increase in DNA methylation at TCF3-ICR region in MCF7 cells, normoxia versus hypoxia. **H)** DNMT3A ChIP qPCR showing change in DNMT3A binding at TCF3-ICR region in MCF7 cells treated with DMSO (Nx versus Hx) or 5 μM GSK591(Hx). **I)** MeDIP qPCR showing decrease in DNA methylation at TCF3-ICR region in MCF7 cells after treatment with DMSO versus 5 μM GSK591under hypoxia. **J)** MeCP2 ChIP qPCR depicting change in MeCP2 binding at TCF3-ICR region in MCF7 cells treated with DMSO (Nx versus Hx) or 5 μM GSK591(Hx). **K)** Immunoblot showing decrease in MeCP2 protein levels in shMECP2 MCF7 cells subjected to hypoxia. **L)** qRT-PCR showing increase in exon 18A/18B ratio in shMECP2 MCF7 cells under hypoxia. Error bars, mean ± SEM; two-tailed *t* test, one-way ANOVA. *$p < 0.05$, **$p < 0.01$, ***$p < 0.001$, ****$p < 0.0001$, $n = 3$ biological replicates. Numerical data of (A), (E), (G–J), (L) available in S1 Data, sheet "Figure S4."
(DOCX)

**S5 Fig. PRMT5-mediated changes at the ICR region is necessary for *TCF3* alternative splicing. A)** RNA pol II ChIP qPCR demonstrating change in RNA pol II occupancy at TCF3-ICR region in MDA-MB-231 cells treated with DMSO (Nx versus Hx) or 5 μM GSK591(Hx). **B)** SpliceAid image showing binding of different splicing factors at the TCF3-ICR RNA. **C)** Immunoblot showing reduction in PTBP1 protein level upon knock down of PTBP1 in MCF7 and MDA-MB-231 cells under hypoxia. **D)** Immunoblot showing PTBP1 in PTBP1 pulldown upon PAR-CLIP. **E)** Sanger sequencing traces analyzed in DECODR showing CRISPR/Cas9 edited sequences. **F)** Plot from DECODR showing Indel distribution. **G)** Matrigel invasion assay and its respective quantification upon pCMV-3tag-1A- EV versus TCF3-18A versus TCF3-18B overexpression in MDA-MB-231 cells. Error bars, mean ± SEM; two-tailed *t* test, one-way ANOVA. *$p < 0.05$, **$p < 0.01$, ***$p < 0.001$, ****$p < 0.0001$, $n = 3$ biological replicates. Numerical data of (A) available in S1 Data, sheet *"Figure S5."*
(DOCX)

**S6 Fig. TCF3-18B (E47) isoform promotes invasion of breast cancer cells. A)** Immunoblot showing reduction in E-Cad expression upon TCF3-18B isoform overexpression in MCF7 cells. **B)** Immunoblot showing over expression of TCF3 18A or TCF3-18B isoform detected using anti FLAG antibody. **C)** Chromatogram depicting TCF3 exon-18A and exon-18B over expression construct cloned in pCMV-3tag1A vector. **D)** Matrigel invasion assay and its respective quantification upon pCMV-3tag-1A-EV versus TCF3-18A versus TCF3-18B overexpression in MDA-MB-231 and MCF7 cells. **E)** Matrigel invasion assay and its respective quantification in shPRMT5 MCF7 cells upon transfection of dCAS9-PRMT5-EV or dCAS9-PRMT5-sgICR under hypoxia. Scale bar, 2 μm. Error bars, mean ± SEM; two-tailed *t* test, one-way ANOVA. *$p < 0.05$, **$p < 0.01$, ***$p < 0.001$, ****$p < 0.0001$, $n = 3$ biological replicates. Numerical data of (D–E) available in S1 Data, sheet *"Figure S6."*
(DOCX)

**S1 File. Sheet 1:** EMT TFs list. **Sheet 2:** List of genes for GO analysis. **Sheet 3:** Genes in the highlighted GO category.
(XLSX)

## Acknowledgments

The authors thank all the members of ERPL for their immense support.

## Author contributions

**Conceptualization:** Srinivas Abhishek Mutnuru, Pooja Yadav, Sanjeev Shukla.

**Data curation:** Srinivas Abhishek Mutnuru.

**Formal analysis:** Srinivas Abhishek Mutnuru.

**Funding acquisition:** Sanjeev Shukla.

**Investigation:** Srinivas Abhishek Mutnuru, Parik Kakani, Shruti Ganesh Dhamdhere, Poorva Kumari.

**Methodology:** Srinivas Abhishek Mutnuru, Pooja Yadav, Parik Kakani, Shruti Ganesh Dhamdhere, Poorva Kumari.

**Project administration:** Srinivas Abhishek Mutnuru, Sanjeev Shukla.

**Resources:** Shruti Agrawal, Atul Samaiya.

**Supervision:** Sanjeev Shukla.

**Validation:** Srinivas Abhishek Mutnuru.

**Visualization:** Srinivas Abhishek Mutnuru.

**Writing – original draft:** Srinivas Abhishek Mutnuru.

**Writing – review & editing:** Srinivas Abhishek Mutnuru, Sanjeev Shukla.

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
