## [Editor Report · Decision Letter 0]

30 Nov 2024

Dear Dr Shukla,

Thank you for submitting your manuscript entitled "PRMT5 mediated histone methylation regulates alternative splicing of TCF3 via MECP2-PTBP1 to promote EMT in breast cancer hypoxia" for consideration as a Research Article by PLOS Biology. Please accept my apologies for the delay in getting back to you as we consulted with an academic editor about your submission.

Your manuscript has now been evaluated by the PLOS Biology editorial staff, as well as by an academic editor with relevant expertise, and I am writing to let you know that we would like to send your submission out for external peer review.

Once your full submission is complete, your paper will undergo a series of checks in preparation for peer review. After your manuscript has passed the checks it will be sent out for review. To provide the metadata for your submission, please Login to Editorial Manager (https://www.editorialmanager.com/pbiology) within two working days, i.e. by Dec 02 2024 11:59PM.

Kind regards,

Richard

Richard Hodge, PhD

rhodge@plos.org

PLOS

---

## [Decision Letter · Decision Letter 1]

20 Jan 2025

Dear Dr Shukla,

Thank you for your patience while your manuscript "PRMT5 mediated histone methylation regulates alternative splicing of TCF3 via MECP2-PTBP1 to promote EMT in breast cancer hypoxia" was peer-reviewed at PLOS Biology. Please accept my sincere apologies for the delays that you have experienced during the peer review process. Your manuscript has been evaluated by the PLOS Biology editors, an Academic Editor with relevant expertise, and by three independent reviewers.

As you will see in the reviewer reports, which can be found at the end of this email, although the reviewers find the work potentially interesting, they have also raised a substantial number of important concerns. Based on their specific comments and following discussion with the Academic Editor, it is clear that a substantial amount of work would be required to meet the criteria for publication in PLOS Biology. However, given our and the reviewer interest in your study, we would be open to inviting a comprehensive revision of the study that thoroughly addresses all the reviewers' comments. Given the extent of revision that would be needed, we cannot make a decision about publication until we have seen the revised manuscript and your response to the reviewers' comments. Your revised manuscript would need to be seen by the reviewers again, but please note that we would not engage them unless their main concerns have been addressed.

The reviewers are generally interested in the findings and the model proposed, but each raise slightly different and overlapping concerns. The reviewers note that several controls under normoxic conditions are missing and they ask that normoxic experiments should be included for each set of hypoxic assays. In addition, Reviewer #1 asks that several experiments are included to provide more direct evidence for the model presented in Figure 7 and raises concerns with the use of a PRKMT5 inhibitor that can have pleiotropic effects. Finally, Reviewer #3 asks whether hnRNP H1 varies upon hypoxia and contributes to TCF3 splicing and notes that the two breast cancer lines used have very different properties even though they are treated as equivalent EMT models.

We appreciate that these requests represent a great deal of extra work, and we are willing to relax our standard revision time to allow you 6 months to revise your study. Please email us (plosbiology@plos.org) if you have any questions or concerns, or envision needing a (short) extension.

**IMPORTANT - SUBMITTING YOUR REVISION**

*Resubmission Checklist*

*Published Peer Review*

*PLOS Data Policy*

*Blot and Gel Data Policy*

Best regards,

Richard

Richard Hodge, PhD

rhodge@plos.org

REVIEWS:

Reviewer #1: In this paper, the authors propose that PRMT5 is upregulated in hypoxic conditions and that it controls the splicing of TCF3 increasing the expression of the E47 isoform that contains Exon 18B that promotes a more invasive phenotype. Mechanistically, they propose that under hypoxic conditions, CTCF binds to the PRMT5 promoter leading to increased transcription, PRMT5 increases the methylation of H4R3 and H3R8 in nucleosomes that are within a regulatory intronic region of TCF3 gene that enables the recruitment of DNMT3A and DNA methylation, the recruitment of the methyl-reader MeCP2 that stalls RNA PolII leading to recruitment of the splicing factor PTBP1.

This is in an interesting paper as the mechanisms by which PRMT5 regulates splicing independent of spliceosome assembly is largely unknown, and there is a large amount of data and some nice use of the dCAS9 system for loci-specific protein recruitment. However, I am not convinced that the data is fully supportive of the model they propose in Figure 7, particularly the link between histone symmetric dimethylation at the TCF3 intronic region and alteration in splicing. As such, in many places the results are over-interpretated. The authors repeatedly state that an effect induced by the PRMT5 inhibitor GSK591 is evidence that histone methylation is the mechanism which PRMT5 regulates splicing at the TCF3 locus, however this cannot be said given that all PRMT5 substrate methylation, including that of splicing factors, will be reduced. This is a major concern as it alters the interpretation of the data and the model.

Moreover, whilst the data in Figure 1 is convincing, the literature already supports a role for PRMT5 in hypoxia (PRMT5 promotes tumour growth and the methylation of ULK1 in hypoxic conditions (PMID: 35246531); hypoxic treatment of human lung cells increases PRMT1 and PRMT5 expression (PMID: 24121109), PRMT5 expression is upregulated in hypoxic endothelial cells (PMID: 35531958)), which was not discussed in the introduction.

This paper has been hard to make comments due to a lack of page and line numbers. Please can this be included in revision.

Major corrections:

1. Data in 1A that concludes that PRMT5 is the main PRMT whose expression is upregulated in hypoxia is not convincing, possibly because of a change of scale used between the normoxic and hypoxic sections. As such, it unclear how data in 1B is generated.

2. I have major concerns for the H4R3me2 and H3R8me2s WB and ChIP data generated in this study as it is well known that these histone marks are expressed at very low abundance and that antibodies are not well characterised. The full WB for 1E need to be included in supplementary information and controls (GSK591) such as those shown in Figure 3C but using a lower dose of GSK591 (<1uM) as 5uM is very high. Likewise, inclusion of PRMT5 knockdown in H4R3me2s and H3R8me2s ChIP (Figure 5B) are important controls to draw conclusions.

3. Since PRMT5 has already been associated with EMT-associated gene changes in normoxic conditions, can the authors compare the extent in changes in CDH1, CDH2, VIM and SNAI1 transcripts after PRMT5 knockdown/inhibition in hypoxic and normoxic conditions (Figure 3A, 3D).

4. Likewise, does knockdown of CTCF affect PRMT5 levels in normoxic conditions (Figure 2D). How can the authors be sure that the effects of CTCF KD on PRMT5 expression is not due to global disruption of gene expression given the insulator role for CTCF?

5. The authors conclude because GSK591 changes the expression of EMT genes (Figure 3D/E) that "PRMT5-mediated histone modifications were involved in regulation of EMT induction after hypoxia". Given that GSK591 will affect the methylation of splicing factors and other proteins involved in transcription in addition to histones, this statement cannot be made and is misleading.

6. ΔPSI is normally set to a 10% change. The data should be re-analysed using these thresholds as 3% change is very low. Sashimi plots for TCF3, ZEB2 and ZBTB38 should be included to show quantitatively the change in exon usage.

7. Figure 4: There is no indication the number of significant spliced events (ΔPSI), rather just the classification of events. This needs to be included, along with the list of genes present in the highlighted GO categories. Figure 4I is not controlled, a control siRNA should be included.

8. Figure 5A that depicts the position of the ICR should be amended to include the length of the ICR and it position from slice junctions.

9. Again, a conceptual jump made in the text that is not supported by the data: "We observed a significant decrease in 18A/18B ratio upon targeting dCAS9PRMT5 to the TCF3 locus (Figure 5E) indicating that local histone R dimethylation cataslyed by PRMT5 at the TCF3 splicing locus is sufficient to alter the alternative splicing pattern of TCF3". I agree that hypoxia and dCAS9PRMT5 increases histone methylation at the ICR, however the link to 18A/B isoform switch is at best correlative. Mutation of H3/H4 at the ICR would be required to support histone methylation regulating splicing changes. In line with this, the authors state that "We also observed that abrogation of the histone marks is sufficient to inhibit DNMT3A recruitment and DNA methylation at the TCF3 ICR region" (text after signpost for figure 5H), however I fail to see this data. If the authors are suggesting that GSK591 treatment is evidence to support this then I do not agree given that all PRMT5 methylation will be inhibited. There is also no evidence that PRMT5-mediated histone methylation impairs RNA PolII binding as again the authors are using GSK591 (Figure 6A), nor that PTBP1 recruitment is dependent on PRMT5-histone methylation (Figure 6D).

10. The RNA PolII ChIP experiments (Figure 6A, B) are only showing binding of RNA PolII at that locus and thus I struggle to see how the data supports alterations in RNA PolII elongation rates. As such, there is no data supporting that "MeCP2 binding leads to slower RNA PolII elongation rates".

11. Data in Figure 6F is not convincing.

Minor corrections:

1. Antibodies need to be stated if used for ChIP, IHC-F or WB and what dilution was used.

2. No indication of how long GSK591 is added to cell culture media before analysis.

Reviewer #2: Within the manuscript "PRMT5 mediated histone methylation regulates alternative splicing of TCF3 via MECP2-PTBP1 to promote EMT in breast cancer hypoxia" by Mutnuru et al., the authors find that the arginine methylatransferase PRMT5 is upregulated under hypoxic conditions, dependent on displacement of CTCF. The elevation of PRMT5, in turn, impacts the expression of EMT related proteins, in general elevating their expression and regulating alternative splicing events. In particular, the alternative splicing of TCF3, as controlled by PRMT5, is studied is some detail.

Some impressive results are found within the manuscript. For example, that targeting dCAS9-PRMT5 to the TCF locus leads to alternative splicing is quite novel. This is a robust approach and informs of the importance of PRMT5 activity in regulating gene splicing. The team also found that knockdown of PRMT5 leads to localized loss of DNA methylation, strengthening a relationship between PRMT5 and DNA methylation. That the alternative spliced variant of TCF3 (18B) has enhanced abilities to promote matrigel invasion is also quite interesting.

A principal conclusion of the paper is that PRMT5 plays a unique role under conditions of hypoxia. However, many controls under normoxic conditions are missing, weakening this point. For each set of experiments carried our under hypoxic conditions, a set carried out under normoxic should be included as a control.

Major points:

1. Many critical "normxic" controls are missing. This is especially true for experiments carried out under hypoxic conditions, where normoxic controls are frequently missing. It is impossible to conclude that certain PRMT5 activities are specific to hypoxic conditions if normoxic controls have not been considered.

Missing controls include:

i) Figure 2I. dCAS9 targeting CTCF binding site under hypoxic conditions lead to increased DNA methylation.

- This experiment should be carried out under normoxic conditions as a control.

ii) Figure 3F. The impact of shPRMT5 on viability under normoxic conditions should be examined as a control. PRMT5 is known to be important for viability in general. Is there a difference between normoxic and hypoxic conditions ?

iii) Similarly to pint (ii), is there a difference in response of 5μM GSK591 under hypoxia versus normoxic conditions.

iv) Hongshan Chen et al., Oncogene 2016 showed that PRMT5 knockdown under basal conditions reduces invasion. How is the current data showing viability of PRMT5 knockdown novel?

-It is unclear that there are differences in viability or invasion between PRMT5 knockdown under normoxic or hypoxic conditions. This needs to be explored.

v) That TCF3 splicing is controlled by DNMT and PRMT5 under hypoxic conditions requires further controls. The knockdown of these enzymes should be carried out under normoxic conditions as a comparison. Likewise, shRNA-mediated knockdown of PTBP1 and its impacts should be compared under hypoxic and normoxic conditions.

2. Other important controls are missing.

i) In figure 1E it is shown that histone Arginine methylation increases under hypoxic conditions. An important control is to assess histone non-arginine methylation, such as H3K9me3, and H3K4me3. Do all methylation events change? Or is this specific to arginine methylation?

ii) Figure 2F. Me-DIP controls are missing. Other CTCF-bound promoters should be examined as well as non-CTCF bound promoters.

iii) Figure 6a. For ChIP-qPCR it is important to use primers targeting non-transcribed regions of the genome as negative controls. This is standard.

3. The authors claim that PRMT5 regulates EMT-related factors via splicing events. However, Lei Huang et al, Nat Comm 2022 demonstrate that PRMT5 regulates EMT factors, especially TCF3, via translational regulation.

- How can these differences be resolved?

4. A major shortcoming of the paper is that alternative splicing on the level of protein expression has not been examined.

Can the alternative splicing of TCF under hypoxic conditions be confirmed by Western blotting or other proteomic approaches ?

Many studies support splicing variants under various conditions but it is rarely proven on the level of protein expression.

A Western blot of TCF using an antibody recognizing an antigen outside the spliced region under hypoxic and normoxic conditions would be insightful.

5. The data shown in Figure 1A are challenging to interpret. It appears that all of the PRMTs remain fairly constant across both normoxic and hypoxic conditions. Only PRMT 12 seems to undergo a change in mRNA levels. Can these data be combined to a single graph showing the normoxic to Hypoxic ratio?

What does "significantly upregulated mean"? What is the level of upregulation quantified?

However, the conclusion that PRMT5 is upregulated under hypoxic conditions compared to normoxic seems to be amiss.

Reviewer #3: This paper aims to identify mechanisms through which PRMT5 influences hypoxia induce alternative splicing. A detailed series of experiments are performed demonstrating (1) That CTCF induces PRMT5 expression through a DNA methylation sensitive promoter element, (2) PRMT5 controls features of EMT in breast cancer cell lines, (3) PRMT5 regulates alternative splicing of TCF3 via DNA methylation dependent recruitment of a complex of factors (symmetric arginine demethylation, DNMT3A, MeCP2) resulting in RNA polII pausing and PTBP1 recruitment to a intronic conserved region (IRC). While hypoxia, PRMT5 and TCF3 splicing are known to be involved in EMT, this study nicely links these factors and provides another example demonstrating a link between histone modifications and their impact of alternative splicing.

Major comments

(1) This focus of regulator activity for TCF3 splicing appears to be the ICR, a region previously shown to be regulated be bound by PTBP1 (PMID: 31391218). In this paper Yamazaki et al propose hnRNP H1 levels to be a major determinant of TCF3 splicing outcome. Do the levels of hnRNP H1 vary upon hypoxia and contribute to TCF3 splicing? The majority of inferences are drawn for ChIP or CLIP qPCR analysis of the ICR region. Deletion of the genomic ICR or mutation of its sites of DNA methylation would provide stronger evidence of a link between hypoxia, PRMT5 and downstream mechanisms involved in TCF3 splicing.

(2) Two cell lines are used and treated as equivalent EMT models although are very different in their properties. MCF-7 cells are epithelial while MDA-MB-231 cells are mesenchymal and are at opposing ends of the EMT spectrum. Does hypoxia induce EMT features within MCF-7 cells, allowing them to become more invasive? What is the relative amount of E47 to E12 within each cell line and how do they differ between cell lines under normoxia and hypoxia? Measurements of splicing in the paper are performed with qPCR and therefore differences in the levels of individual isoforms are difficult to assess. Does PRMT5 manipulation induce a large shift in percent spliced in (PSI) from E12 to E47 TCF3 splicing consistent with potential roles for this splice change in influencing EMT?

Minor comments

(1) The TCF3 splicing outcome is listed as a skipped exon event but is actually a mutually exclusive splicing event.

(2) Fig 4F could be moved to Supplementary.

(3) Although not investigated, the authors could discuss the potential differential functions of E47 and E12 with regard to their impact on EMT.

---

## [Decision Letter · Decision Letter 2]

29 Sep 2025

Dear Dr Shukla,

Thank you for your patience while we considered your revised manuscript "PRMT5 mediated histone methylation regulates alternative splicing of TCF3 via MECP2-PTBP1 to promote EMT in breast cancer hypoxia" for publication as a Research Article at PLOS Biology. Please accept my sincere apologies for the delays that you have experienced during this round of the peer review process. This revised version of your manuscript has been evaluated by the PLOS Biology editors, the Academic Editor and the original reviewers.

Based on the reviews, I am pleased to say we are likely to accept this manuscript for publication, provided you satisfactorily address the remaining points raised by the reviewers. After discussions with the Academic Editor, we will not make the request to demonstrate conversion of the splicing pattern towards a more mesenchymal profile essential for the revision. However, we do ask that the requested discussions/data included in the rebuttal are added to the main manuscript.

In addition, please also make sure to address the following data and other policy-related requests that I have provided below (A-I):

(A) We routinely suggest changes to titles to ensure maximum accessibility for a broad, non-specialist readership. In this case, we would suggest a minor edit to the title, as follows. Please ensure you change both the manuscript file and the online submission system, as they need to match for final acceptance:

“PRMT5 regulates alternative splicing of TCF3 under hypoxia to promote EMT and invasion in breast cancer”

(B) In the human ethics statement in the Methods, please specify whether the consent you obtained from the patients to conduct the study was informed.

(C) You may be aware of the PLOS Data Policy, which requires that all data be made available without restriction: http://journals.plos.org/plosbiology/s/data-availability. For more information, please also see this editorial: http://dx.doi.org/10.1371/journal.pbio.1001797

-Supplementary files (e.g., excel). Please ensure that all data files are uploaded as 'Supporting Information' and are invariably referred to (in the manuscript, figure legends, and the Description field when uploading your files) using the following format verbatim: S1 Data, S2 Data, etc. Multiple panels of a single or even several figures can be included as multiple sheets in one excel file that is saved using exactly the following convention: S1_Data.xlsx (using an underscore).

-Deposition in a publicly available repository. Please also provide the accession code or a reviewer link so that we may view your data before publication.

Figure 1A-D, 1F, 1I, 2B, 2F-G, 2I-J, 2L, 3A, 3D, 3F-G, 4A-B, 4F, 4H-I, 5B, 5D-L, 6A-G, 6I, S2A, S2E-F, S2H, S3A, S3E, S3H-I, S4A, S4E, S4G-J, SKL, S5A, S6D-E

(D) Thank you for providing the RNA-sequencing data in the GEO database (GSE279474). However, we note that the data is currently on hold for release. We ask that you please make the structures publicly available at this stage before publication.

(E) Please also ensure that each of the relevant figure legends in your manuscript include information on *WHERE THE UNDERLYING DATA CAN BE FOUND*, and ensure your supplemental data file/s has a legend.

(F) We require the original, uncropped and minimally adjusted images supporting all blot and gel results reported in the following Figures:

Figure 1E, 1G, 2D, 2K, 3B-C, 3E, 4G, S2D, S3B-D, S3F, S4B, S4D, S4K, S5C-D, S6A-B

We will require these files before a manuscript can be accepted so please prepare and upload them now. Please carefully read our guidelines for how to prepare and upload this data: https://journals.plos.org/plosbiology/s/figures#loc-blot-and-gel-reporting-requirements. *We note that Figure S1 looks to be provide the raw blots for Figure 1E, but we ask that the raw and uncropped images are provided in a separate raw image file in the Supplementary Information.

(G) Please ensure that your Data Statement in the submission system accurately describes where your data can be found and is in final format, as it will be published as written there.

(H) Please ensure that you are using best practice for statistical reporting and data presentation. These are our guidelines https://journals.plos.org/plosbiology/s/best-practices-in-research-reporting#loc-statistical-reporting and a useful resource on data presentation https://journals.plos.org/plosbiology/article?id=10.1371/journal.pbio.1002128

- If you are reporting experiments where n ≤ 5, please plot each individual data point.

(I) Please note that per journal policy, the model system/species studied should be clearly stated in the abstract of your manuscript.

We expect to receive your revised manuscript within two weeks.

*Published Peer Review History*

*Press*

Best regards,

Richard

Richard Hodge, PhD

rhodge@plos.org

Reviewer remarks:

Reviewer #1: I thank the authors for conducting an excellent good job of addressing the comments, and the manuscript is much improved. I am supportive of publication.

Reviewer #2: In the revised manuscript titled "PRMT5 mediated histone methylation regulates alternative splicing of TCF3 via MECP2-PTBP1 to promote EMT in breast cancer hypoxia" the authors have provided experiments that have strengthened their conclusion that PRMT5 acts under hypoxic conditions to generate a splice variant of TCF4.

Specifically, the authors have carried out control experiments under normoxic conditions using a CTCF-binding site targeting dCAS9 construct,examined viability after shPRMT knockdown under normoxic conditions, a shRNA-mediated knockdown of PTBP1 under hypoxic and normoxic conditions, an expansion of histone modification controls, and CTCF binding site controls. Importantly, a Western blot, supporting alternative splicing of TCF4 under hypoxic conditions has also been provided.

In sum, all of the requested revisions have been satisfactorily carried out and the manuscript is now ready for publication.

Reviewer #3 (Philip Gregory, identifies himself): The authors have included new data, deleting the ICR region, and shown it influences TCF3 splicing under hypoxia. This provides stronger evidence that the ICR mediates splicing. The additional data presented in Rebuttal Figure 22 does not provide comparative measurements of TCF3 splicing between MCF-7 and MDA-MB-231 cells, only relative changes within each cell line. Although not essential, it is important to note that MCF-7 and MDA-MB-231 cells represent epithelial and mesenchymal cell types and it would be interesting to see whether hypoxia shifts the TCF3 splicing profile in MCF-7 cells to a more mesenchymal profile - as would be expected to be observed in MDA-MB-231 cells. It would be beneficial to include some of the author's additional discussion and data presented as rebuttal figure 20 and in response to minor comment (3) within the Discussion of this manuscript.

---

## [Editor Report · Decision Letter 3]

2 Oct 2025

Dear Dr Shukla,

On behalf of my colleagues and the Academic Editor, Tom Misteli, I am pleased to say that we can in principle accept your manuscript for publication, provided you address any remaining formatting and reporting issues. These will be detailed in an email you should receive within 2-3 business days from our colleagues in the journal operations team; no action is required from you until then. Please note that we will not be able to formally accept your manuscript and schedule it for publication until you have completed any requested changes.

In addition, I have suggested a minor edit to the Abstract for flow and language below my signature. I would be grateful if this could be incorporated during the production process if you agree with the changes.

PRESS

Best wishes, 

Richard

Richard Hodge, PhD

rhodge@plos.org

PLOS

--EDITED ABSTRACT--

Tumor hypoxia induced alterations in the epigenetic landscape and alternative splicing influence cellular adaptations. PRMT5 is a type II protein arginine methyltransferase that regulates several tumorigenic events in many cancer types. However, the regulation of PRMT5 and its direct implication on aberrant alternative splicing under hypoxia remains unexplored. In this study, we observed hypoxia induced upregulation of PRMT5 via the CTCF in human breast cancer cells. Further, PRMT5-mediated symmetric arginine dimethylation H4R3me2s and H3R8me2s directly regulated the alternative splicing of TCF3. Under hypoxia, PRMT5-mediated histone dimethylation at the intronic conserved region (ICR) present between TCF3 exon 18a and exon 18b recruits DNMT3A, resulting in DNA methylation. DNA methylation at the TCF3-ICR is recognized and bound by MeCP2 resulting in RNA-Pol II pausing, promoting the recruitment of the negative splicing factor PTBP1 to the splicing locus of TCF3 mRNA. PTBP1 promotes the exclusion of exon 18a which results in the production of the proinvasive TCF3-18B (E47) isoform which promotes EMT and invasion of breast cancer cells under hypoxia. Collectively, our results indicate PRMT5-mediated symmetric arginine dimethylation of histones regulates alternative splicing of TCF3 gene thereby enhancing EMT and invasion in breast cancer hypoxia.